# The impacts of climate change on tropical-to-extratropical transitions in the North-Atlantic basin

Aude Garin[1], Francesco S.R. Pausata[1], Mathieu Boudreault[2] Roberto Ingrosso[1]

[1]Department of Earth and Atmospheric Sciences, Université du Québec à Montréal, Montréal, Québec, Canada

[2]Department of Mathematics, Université du Québec à Montréal, Montréal, Québec, Canada

*Correspondence to*: Aude Garin (garin.aude@courrier.uqam.ca)

**Abstract.** As tropical cyclones migrate towards mid-latitudes, they can transform into extratropical cyclones, a process known as extratropical transition. In the North Atlantic basin, nearly half of the hurricanes undergo this transition. After transitioning, these storms can reintensify, posing significant threats to populations and infrastructure along the eastern coast of North America. While the impacts of climate change on hurricanes have been extensively studied, there remain uncertainties about its effects on extratropical transitions. This study aims to assess how climate change affects the frequency, location, intensity, and duration of these transitions. To achieve this, high-resolution regional simulations from an atmospheric regional climate model, based on the RCP 8.5 emissions scenario, were used to compare two 30-year periods: the present (1990-2019) and the end of the century (2071-2100). The results indicate a projected decrease in the number of tropical hurricanes, with no significant change in extratropical transition rates. September and October continue to be the primary months for extratropical transitions. However, the season's peak appears to have shifted from September to October, suggesting that large-scale environmental conditions may become more favorable for extratropical transitions in October in the future. Although a poleward shift in the maximum intensity of tropical hurricanes is detected, the average latitude of the transitions does not change. Our findings suggest that transitioning storms will be more intense in the future, despite a less baroclinic atmosphere due to a stronger contribution from latent heat transfer. However, the risk of reintensification after transition is not expected to increase.

## 1. INTRODUCTION

Tropical cyclones can transform into extratropical cyclones through a process called extratropical transition (ET), in particular when they encounter a baroclinic environment and experience cooler sea surface temperatures. These transitions occur rather frequently in the North Atlantic basin, with approximately 50% of hurricanes undergoing an ET (Bieli et al., 2019; Hart & Evans, 2001). Moreover, about 50% of tropical cyclones that made landfall during this period were in the process of transitioning (Hart & Evans, 2001). Some of these cyclones can intensify after transitioning, resulting in significant harm to human lives and damage to infrastructures, as seen with Hurricane Floyd (1999) and Hurricane Sandy (2012). More recently, Hurricane Fiona in 2023 has set a new record for the lowest pressure ever recorded in Canada (Chedabucto Bay). The North-Eastern Coast of the United States and the Canadian Maritimes, which typically experience 1-2 of these storms per year, as well as Western Europe, which faces them once every two years, are particularly vulnerable to these risks (Hart & Evans, 2001). Given the substantial financial losses these events can cause, there are growing concerns among economic stakeholders about the potential impact of climate change on the frequency and severity of these transitions.

The atmospheric mechanisms underlying extratropical transitions have been extensively studied in recent years. As a tropical cyclone moves poleward, it encounters several environmental changes, such as low sea surface temperature (SST), increased SST gradients, enhanced vertical wind shear, and a stronger Coriolis

force (Jones et al., 2003). These changes may remarkably affect the tropical cyclone, resulting in a loss of
intensity, a breakdown of its warm-core structure, and a loss of its axisymmetric structure (Jones et al., 2003).
When a tropical cyclone encounters an existing extratropical system, typically a mid-tropospheric trough,
the process of extratropical transition may be initiated (Arnott et al., 2004; Hart et al., 2006; Jones et al.,
2003; Klein et al., 2000; Wood & Ritchie, 2014). The mid-tropospheric trough favors the advection of the
angular momentum – rather than the heat advection – which drives the conversion of the cyclone's warm-
core structure into a cold-core structure (Hart et al., 2006). This advection of angular momentum disturbs the
classic structure of a tropical cyclone, characterized by a decrease in wind strength with height and a warm
core, thereby disturbing the thermal wind balance. To restore thermal wind balance, a secondary circulation
is established: above the maximum of angular momentum advection, adiabatic descent causes warming,
while below the maximum, adiabatic ascent leads to cooling, resulting in frontogenesis (Harr & Elsberry,
2000; Hart et al., 2006). During this transition, the storm's wind field expands and becomes asymmetric,
shifting the location of maximum wind speeds (Evans & Hart, 2008). The timing of the transition from a
warm-core to an asymmetric structure appears to be basin-dependent: in the North Atlantic basin, TCs seem
to acquire first an asymmetric structure while maintaining a warm-core structure, whereas, in the Eastern
North Pacific they first lose their warm-core structure while retaining their symmetry (Bieli et al., 2019;
Wood & Ritchie, 2014).
Hence, the mid-latitude environment and weather systems play an essential role in a tropical cyclone's
transition process. During and after the extratropical transition, the addition of baroclinic energy (Evans et
al., 2017) and diabatic heating (Rantanen et al., 2020) may cause an intensification of post-transition tropical
cyclones. The structure of the tropical cyclone may play a role in the reintensification process after ET (Arnott
et al., 2004) as well as the presence of atmospheric rivers which may speed up the process (Baatsen et al.,
66   2015).

While the impact of climate change on tropical cyclones has been largely addressed in recent years, only a
few studies have focused on the effects of a warmer climate on extratropical transition events. Regarding
tropical cyclones, it is expected that their global frequency may decrease in a warmer environment, although
the proportion of very intense hurricanes (especially Category 4 and 5 events) is likely to increase (Bender
et al., 2010; Hill & Lackmann, 2011; Knutson et al., 2020; Mallard et al., 2013). Additionally, there may be
a poleward expansion of tropical cyclone genesis (Garner et al., 2021) along with a poleward migration of
the maximum intensity of tropical cyclones (Lee et al., 2020) due to higher SSTs and reduced wind shear at
mid-latitudes. However, there remain uncertainties around this latter point (Knutson et al., 2020). In the mid-
latitude environment, a decrease in vertical wind shear (Kossin et al., 2014) and a reduction in the frequency
of extratropical systems during the summer season in the Northern Hemisphere (Lehmann et al., 2014) are
anticipated. Consequently, storms might become more intense during extratropical transition due to the
projected increase in the intensity of tropical cyclones. Furthermore, extratropical transitions may occur
farther north. Warmer SSTs could enable tropical cyclones to maintain their tropical characteristics and
strength further north, thereby increasing the likelihood of encountering a baroclinic zone necessary for
extratropical transition (Hart & Evans, 2001). However, the potential projected decrease in extratropical
systems during summer (Lehmann et al., 2014), along with the reduction in baroclinicity in the low
troposphere, may hinder extratropical transitions.
Regarding extratropical transitions in a warmer environment, they are expected to be more intense, last
longer, and be associated with heavier precipitation (Jung & Lackmann, 2019, 2021, 2023; Michaelis &
Lackmann, 2019, 2021). However, post-transition storms are expected to be less intense (Jung & Lackmann,
2021, 2023). An increase in the frequency of extratropical transition events is anticipated in the central and
eastern North Atlantic basin (Baker et al., 2022; Liu et al., 2017), consistent with the projected poleward and
eastward expansion of tropical cyclone genesis regions. This shift in the genesis location may likely lead to
more storms hitting Western Europe (Baatsen et al., 2015; Haarsma et al., 2013). However, there is still
uncertainty about how extratropical transition events will evolve in a warmer environment.
Our study aims to determine the impacts of climate change on the frequency, location, duration and intensity
of extratropical transitions in the North Atlantic basin by using high-resolution simulations from a regional
atmospheric climate model that span two 30-year periods: 1990-2019 for present-day simulations and 2071-
2100 for simulations of the future. Simulations of future climate are based on the Representative
Concentration Pathways 8.5 (RCP 8.5) scenario. The focus of the study is particularly on highly populated
areas (U.S. Northeastern Coast and Canadian Maritimes) where casualties and infrastructure damages can be
significant.
This paper is organized as follows: Section 2 describes the data and methodology used to track tropical
cyclones and extratropical transition events, as well as the key metrics used to assess the change in
extratropical transitions. Section 3 presents the key findings on how climate change impacts the frequency,
location, and intensity of ET. Section 4 discusses these findings and provide conclusions.
**2.  EXPERIMENTS AND METHODS**
**2.1    Data experiments and model description**
The two 30-year experiments used for this study are part of the set of simulations used in Ingrosso & Pausata
(2024). These experiments encompass the present-day scenario (1990-2019) and the RCP 8.5 future scenario
(2071-2100). We chose to focus on the most extreme scenario to determine whether any significant impacts
emerge, as scenarios with lower greenhouse gas emissions are less likely to produce a discernible signal.
These experiments were performed with the developmental version of the Canadian Regional Climate
Model/Global Environmental Multiscale (CRCM5/GEM 4.8) at a horizontal grid spacing of 0.12° and 57
vertical levels. The regional model CRCM5 was driven using the data from the global simulations performed
with GEM4.8 at 0.55° horizontal resolution and 73 vertical levels (further details can be found in Ingrosso &
Pausata (2024)).
To evaluate the model's performance, the regional model was compared with observations from the Tropical
Rainfall Measuring Mission (TRMM), the Climate Research Unit, and one reanalysis product (ERA5),
focusing on the mean precipitation distribution from 2000 to 2019 (Ingrosso & Pausata, 2024). The regional
model demonstrated its ability to align with the observations despite a persistent dry bias in the median and
lower percentiles. Additionally, the regional model has shown good performance in reproducing the general
diurnal cycle, although rainfall was underestimated compared to satellite observations. The precipitation
comparison is a general evaluation of the model and indeed does not provide meaningful information in terms
of the ability of the model to represent TC/ET climatology which is later discussed in Section 2.8 and in the
Appendix.
The regional area covered by the simulations extends from 3°S to 48°N and from 81°W to 52°E.
**2.2    Storm Tracking algorithm**
In this study, we employ a storm-tracking algorithm designed to detect both tropical cyclones (including
tropical storms) and transitioning tropical cyclones. This algorithm is based on the methodology used in
Dandoy et al. (2021) and follows a three-step procedure: storm identification, storm tracking, and storm
lifetime, in line with previous studies (Gualdi et al., 2008; Scoccimarro et al., 2017). The algorithm uses 3-
hour outputs of the model for the period from June to December.
*Storm identification*
A storm is identified when several criteria are met. One of the key strengths of this algorithm is the double-
filtering approach that prevents from double-counting a tropical cyclone (TC) when there is a temporary
decrease in intensity, followed by a restrengthening. Specifically, each storm center is categorized as either
a weak center (if it meets only loose criteria) or as a strong center (if it also meets strict criteria). The criteria
used are as follows:
-    Surface pressure: The center's surface pressure must be lower than 1013 hPa (1005 hPa to be
classified as a strong center) and represent a minimum within a 250 km radius. Additionally, the
center must be a closed low-pressure system, with the minimum pressure difference between the
center and a circle of grid points in small (400 km) and large (800 km) radii around the center
exceeding 1 and 2 hPa, respectively (4 and 6 hPa to be considered as a strong center).
-    850-hPa vorticity: The maximum 850-hPa vorticity within a 200-km radius around the center must
exceed $10^{-5}$ s$^{-1}$ ($10^{-4}$ s$^{-1}$ to be considered as a strong center).
-    10m wind: The maximum wind speed at 10 meters within a 100-km radius around the center must
exceed 8 m/s (17.5 m/s to be considered as a strong center).

In this study, the criterion based on temperature anomalies was not used to reject centers, thus allowing for
the detection of warm-core storms such as transitioning tropical storms. However, the algorithm still applies
a strict criterion for identifying strong centers. Specifically, this criterion requires that the sum of the
temperature anomalies at 250, 500 and 700 hPa, defined as the difference between the maximum temperature
and the mean temperature within a 200 km radius around the center, must exceed 2°C.
*Storm tracking*
The purpose of this step is to assign each identified center to an existing storm or, if no existing storm can be
linked to the detected center, to consider it as the origin of a new storm. Initially, centers that are more than
250 km apart are treated separately. If two centers are within this distance, only the center with the strongest
vorticity is retained. Storms are tracked using the nearest-neighbor method, a technique also employed in
various studies (Blender et al., 1997; Blender & Schubert, 2000; Schubert et al., 1998). For each existing
storm, the algorithm predicts the potential location of the next center based on the historical trajectory of the
previous two centers. A new center is then assigned to the storm whose predicted location is closest, with
preference given to the nearest center.
*Storm lifetime*
For each determined track, the following final conditions must be met:
-    The lifetime of the storm must exceed 36 hours.
-    The storm must have at least 12 strong centers.
-    The minimum travel distance must be at least 10° of combined latitude and longitude.
-    The number of strong centers must account for at least 77% of the core trajectory, which is defined
as the path between the first and the last strong center.

**2.3     Detection of extratropical transitions (ET)**
The removal of the warm-core loose criterion enables the algorithm to detect both warm-core and cold-core
centers. Simultaneously, the use of the warm-core strict criterion ensures that only storms that have
experienced a tropical cyclone phase are detected.
ET events are identified using the Cyclone Phase Space (CPS) methodology developed by Hart (2003) and
widely employed in previous studies focusing on ET (Baker et al., 2022; Hart et al., 2006; Jung & Lackmann,
2019, 2021, 2023; Liu et al., 2017). This methodology involves three parameters: the lower-tropospheric
thermal axisymmetry of the cyclone ($B$), the lower-tropospheric ($-V_T^L$) and the upper-tropospheric ($-V_T^U$)
thermal winds. These three parameters describe and differentiate the structure of tropical cyclones,
characterized by a warm-core and vertically stacked structure, from that of extratropical cyclones,
characterized by a cold-core and tilted structure.
The use of high-resolution data ensures reliable CPS diagnostics (Hart, 2003).
*Cyclone thermal symmetry (B)*
This parameter allows for identifying the frontal nature of the cyclone, or the absence thereof. It is defined
as the storm-motion-relative 900-600 hPa thickness asymmetry within a 500-km radius (Hart, 2003). For the
Northern Hemisphere, $B$ is defined as:

$$B = \left( \overline{Z_{600\,hPa} - Z_{900\,hPa}}\big|_R - \overline{Z_{600\,hPa} - Z_{900\,hPa}}\big|_L \right) \tag{1}$$

where $Z$ represents the geopotential height, $R$ and $L$ indicate the right and left sides of storm motions, and the
overbar denotes the mean area over a semicircle with a radius of 500 km.
Very low values of $B$ are associated with non-frontal storms such as TCs, while high values of $B$ are
associated with frontal storms such as extratropical cyclones. Hart (2003) suggests that a threshold of 10m is
appropriate for distinguishing non-frontal storms from frontal storms. This threshold has been widely utilized
in other studies focusing on ET (Baker et al., 2022; Hart et al., 2006; Jung & Lackmann, 2019, 2021, 2023;
Liu et al., 2017). However, Zarzycki et al. (2017) indicate that a threshold of 15 m is more appropriate
threshold when using high-resolution.
To calculate the average speed of the 900-600 hPa layer, we considered four sub-layers: 900-850 hPa, 850-
800 hPa, 800-700 hPa, and 700-600 hPa. The mean zonal and meridional speeds for each layer were
computed as follows:

$$\bar{u}_\iota = \sum_{j \in D} u_{i,j} \tag{2}$$

where $D$ represents the 500km-radius circle around the center.
Then, the total mean zonal and meridional speeds are defined as the weighted average of the mean speeds
calculated for each sub-layer:

$$\bar{u} = \sum_{i=1,4} \bar{u}_\iota \omega_i \tag{3}$$

where $\omega_i$ represents the weight of the layer $i$. The weight of each layer is calculated as the ratio of the
difference between the upper-bound pressure and the lower-bound pressure of the layer to the difference
between the upper-bound pressure and the lower-bound pressure of the entire column.
The left layer comprises the points that satisfy the following criteria:

$$\frac{\pi}{180} R\left( lat_{i,j} - lat_{i_0,j_0} \right) > \frac{\pi}{180} R \frac{\bar{v}}{\bar{u}} \cos\left( \frac{\pi}{180} lat_{i,j} \right) \tag{4}$$

and the thickness of the layer is thus defined as:

$$thickness = \frac{1}{N_L} \left( \sum_{k=1}^{N_L} GZ_{600,k} - GZ_{900,k} \right) \tag{5}$$

where $N_L$ is the number of points within the layer.
The same methodology is also applied to the right layer, which is defined as:

$$\frac{\pi}{180} R\left( lat_{i,j} - lat_{i_0,j_0} \right) < \frac{\pi}{180} R \frac{\bar{v}}{\bar{u}} \cos\left( \frac{\pi}{180} lat_{i,j} \right). \tag{6}$$

*Lower and upper thermal winds*
Tropical cyclones are characterized by a decrease in height perturbation with increasing altitude. In contrast,
for extratropical cyclones, the height perturbation decreases with height.
In this study, the height perturbation $\Delta Z$ is calculated as the difference between the maximum geopotential
height and the minimum geopotential height within a 500 km-radius circle ($\Delta Z = Z_{max} - Z_{min}$) and is
proportional to the magnitude of the geostrophic wind ($V_g$)(Hart, 2003)

$$\Delta Z = \frac{dg|V_g|}{f} \qquad (7)$$

where $d$ represents the distance between the two geopotential extrema, $f$ is the Coriolis parameter, and $g$ is
the gravity constant.
Scaled thermal winds can be defined as (Hart, 2003):

$$-V_T^L = \left.\frac{\partial \Delta Z}{\partial ln(p)}\right|_{900\ hPa}^{600\ hPa} \qquad (8)$$

and

$$-V_T^U = \left.\frac{\partial \Delta Z}{\partial ln(p)}\right|_{600\ hPa}^{300\ hPa}. \qquad (9)$$

The lower troposphere corresponds to the 900-600 hPa layer while the upper troposphere corresponds to the
600-300 hPa layer.
A positive value of $-V_T^L$ (i.e. $-V_T^U > 0$) indicates a warm-core structure in the lower (i.e. upper) troposphere,
while a negative value of $-V_T^L$ (i.e. $-V_T^U < 0$) indicates a cold-core structure in the lower (i.e. upper)
troposphere. During ET, the signs of $-V_T^L$ and $-V_T^U$ may differ.
As recommended by Hart (2003), we conducted a linear regression on the vertical profile of $\Delta Z$ to estimate
the thermal wind parameters. The levels used for these regressions are: 900 hPa, 850 hPa, 800 hPa, 700 hPa,
600 hPa, 500 hPa, 400 hPa, and 300 hPa.
*Detection of ET events*
To mitigate the variability in parameters caused by numerical noise, a 12-hour smoothing window is applied,
following the recommendations of Michaelis & Lackmann (2019), who employed a 24-hour smoothing
window. Additionally, a filtering algorithm was employed to exclude highly chaotic trajectories characterized
by multiple transitions. The core principles of this algorithm are:
- Exclusion of ET events occurring below 20° as ET events rarely occur below this threshold (Hart &
Evans, 2001)
- When multiple transitions occur, only the final transition is considered, provided that no reverse
transition follows it.
For this study, the onset of ET is detected when (Liu et al., 2017; Michaelis & Lackmann, 2019):
- $\tilde{B} > 15\ m$ or $-\widetilde{V_T^L} < 0$
Therefore, all TCs that have started an ET are included in our study, including instant-warm seclusions (Sarro
& Evans, 2022), as well as transitioning storms that have not completed their transition within the regional
domain. And the completion of ET is detected when both criteria are simultaneously met.
In other studies (Hart, 2003; Hart et al., 2006; Jung & Lackmann, 2021), the onset of ET was detected when
B exceeded the threshold. However Liu et al. (2017) argued that this methodology might be inadequate for
capturing TCs that transition to cold-core systems before developing an asymmetric structure. Furthermore,
this methodology may also lead to negative ET durations as highlighted in Kitabatake (2011)
**2.4    The Eady Growth Rate: An Indicator of the baroclinicity**
The Eady Growth Rate (EGR) is a widely used indicator of the baroclinicity of the environment (Eady, 1949).
ET are more likely to occur in zones associated with high values of EGR. It is defined as follows (Lindzen
& Farrell, 1980):

$$\sigma = 0.31 \frac{|f|}{N} \left| \frac{dV}{dz} \right| \tag{10}$$

where $f$ is the Coriolis parameter, $\frac{dV}{dz}$ is the vertical wind shear, and $N$ is the Brunt-Väisälä frequency:

$$N = \sqrt{\frac{g}{\theta} \frac{\partial \theta}{\partial z}} \tag{11}$$

where $\theta$ is the virtual potential temperature, and $g$ is the gravity constant.
In this study, we mainly focused on mid-troposphere baroclinicity and, therefore, computed the EGR at 500
hPa with a forward difference scheme, using the geopotential heights, humidity, meridional and zonal wind
speeds, and temperatures at 400 hPa and 500 hPa.
To assess the baroclinicity in the upper troposphere, we computed the EGR at 200 hPa with a backward
difference scheme using the 300 hPa and 200 hPa values. A forward scheme in this case would have required
using the 100 hPa values, introducing stratospheric influences, which we aimed to avoid.
**2.5    Integrated Kinetic Energy: An indicator of the storm intensity**
The concept of Integrated Kinetic Energy (IKE) was first introduced by Powell & Reinhold (2007), who
demonstrated that this indicator might better assess a hurricane's destructive potential than the maximum
sustained surface wind speed, as it accounts for storm size.
IKE is the integration of the 10-m kinetic energy per unit volume over a domain volume ($V$) centered around
the storm's center. IKE is given by:

$$IKE = \int_V \frac{1}{2} \rho U^2 dV \tag{12}$$

where $\rho$ is the air density and $U$ is the 10-m wind velocity.
Assuming an air density value of 1 kg/m$^3$ and a volume height of 1m, the expression can be simplified as
follows (Cheung & Chu, 2023):

$$IKE = \int_A \frac{1}{2} U^2 dA \tag{13}$$

In this study, the area considered is a circle with a 500km radius, centered around the TC center.

## 2.6 Minimal theoretical pressure of a TC

The minimal theoretical pressure allows to estimate the minimum pressure a TC center can reach, based on the SST and the atmospheric profile (Bister & Emanuel, 2002). This critical pressure $p_c$ is given by the following equation:

$$R_d T_s ln\left(\frac{p_a}{p_c}\right) = \frac{1}{2}\left(\frac{T_s}{T_0}\frac{C_k}{C_D}(CAPE^* - CAPE_{env})|_{RMW}\right) + CAPE_{env}|_{RMW} \qquad (14)$$

where $p_a$ is the environmental pressure, $T_s$ is the SST, $T_0$ is the outflow layer temperature, $C_k$ and $C_D$ are the enthalpy and momentum surface exchange coefficient, and $R_d$ is the ideal gas constant for dry air. $CAPE^*|_{RMW}$ is the Convective Available Potential energy of a saturated air parcel and $CAPE_{env}|_{RMW}$ is the environmental Convective Available Potential Energy.

The minimum theoretical pressure was calculated with the pyPI package from Python (Gilford, 2021).

## 2.7 Statistical analysis

For the statistical assessment of the differences, the Mann-Whitney-Wilcoxon test was used to compare the distributions. This test is recommended when the normality assumption cannot be made.

A significance level of 10% was considered.

## 2.8 Validation

The annual ET ratio, defined as the ratio of the yearly number of ET events to the yearly number of tropical cyclones, was computed for the 1990-2019 period of the present-day experiment. This frequency was then compared with observational data from the International Best Track Archive for Climate Stewardship (IBTRACS, Knapp et al., 2010) and with reanalysis data ECMWF reanalysis (ERA5, Hersbach et al., 2020), to which the tracking algorithm was applied within the same spatial domain as the present-day experiment.

The results demonstrate the current experiment's strong ability to reproduce the mean annual ET ratio despite exhibiting a lower distribution variability than ERA5 and IBTRACS (Fig. 1).

Several studies have explored the topic of ET ratio simulation in different basins over the past years using the CPS methodology with different models, resolutions or reanalyses.

Bieli et al. (2019) used JRA-55 and ERA-Interim whereas Zarzycki et al. (2017) used two reanalysis products, ERA-Interim and CFSR, combined with two climate models, CAM-55 et CAM-28. This latter study highlights the importance of the resolution with a 9% increase in the mean annual ET ratio with a higher resolution. Liu et al. (2017) used two reanalysis products, CFRS and JRA-55, combined with two climate models, FLOR et FLOR-FA, for which the SST are artificially corrected through flux adjustment. This correction leads to a better representation of the ET ratio. Studholme et al. (2015) found a very high mean annual ET ratio (68%), this finding being explained by the simulation of longer tracks, enabling the ET to occur.

Table 1 lists the mean annual ET ratios from previous studies. In our study, the ET ratio found in the present-day simulation is 42.7%, placing it at the lower end of the range. However, this value accounts for the adjustments made to the CPS method, as detailed in Section 2.3. Indeed, we noticed that for certain tracks, some storms could begin to acquire extratropical characteristics (asymmetry or a cold core) before reverting to tropical cyclones. These "false" transitions were therefore excluded from the transitions. It is important to point out that if another transition occurs, the storm will be considered among the transitioning storms. Before correcting this, the transition rate was at 68.5% (close to the findings of Studholme et al. (2015)).

| Author(s) | Mean ET fraction | Method/data for tracking ETs |
|---|---|---|
| **Hart & Evans (2001)** | 46% | NHC best track labels |
| **Studholme et al. (2015)** | 68% | CPS and k-means clustering, storms tracked in ECMWF operational analysis |
| **Zarzycki et al. (2016) -1** | 55% | CPS, storms tracked in ERA-Interim |
| **Zarzycki et al. (2016) - 2** | 50% | CPS, storms tracked in CFSR |
| **Zarzycki et al. (2016) - 3** | 49% | CPS, storms tracked in CAM-28 |
| **Zarzycki et al. (2016) - 4** | 40% | CPS, storms tracked in CAM-55 |
| **Liu et al. (2017) - 1** | 56% | CPS, storms tracked in CFRS |
| **Liu et al. (2017) - 2** | 50% | CPS, storms tracked in JRA-55 |
| **Liu et al. (2017) - 3** | 57% | CPS, storms tracked in FLOR-FA |
| **Liu et al. (2017) - 4** | 31% | CPS, storms tracked in FLOR |
| **Bieli et al. (2019) - 1** | 47% | CPS, storms tracked in JRA-55 |
| **Bieli et al. (2019) - 2** | 54% | CPS, storms tracked in ERA-Interim |


**Table 1**: Summary of mean annual ET ratio from previous studies.
ET in IBTrACS is determined subjectively by various forecasters based on real-time observational data. In
addition, IBTrACS phase transition occurs at an instantaneous point in space and time and provides no
information about the path of ET (Zarzycki et al., 2017). To assess the ability of the model to spatially
reproduce ET, we have compared the latitude and the longitude of ET onset with the results of Bieli et al.
(2019) in Table 2. The comparison shows a northward shift in our simulated ET onset latitude compared to
Bieli et al. (2019). This difference may be explained by our methodology, which in the case of multiple
transitions, considers only the final transition.

| Simulation | Mean Latitude ET Onset | Mean Longitude ET Onset |
|---|---|---|
| GEM 4.8/CRMC5 | 35.5 | -52.4 |
| JRA55 (Bieli et al., 2019) | 33.2 | -58.4 |
| ERA5 - Interim (Bieli et al., 2019) | 28.9 | -56.2 |


**Table 2**: ET onset mean latitude and longitude


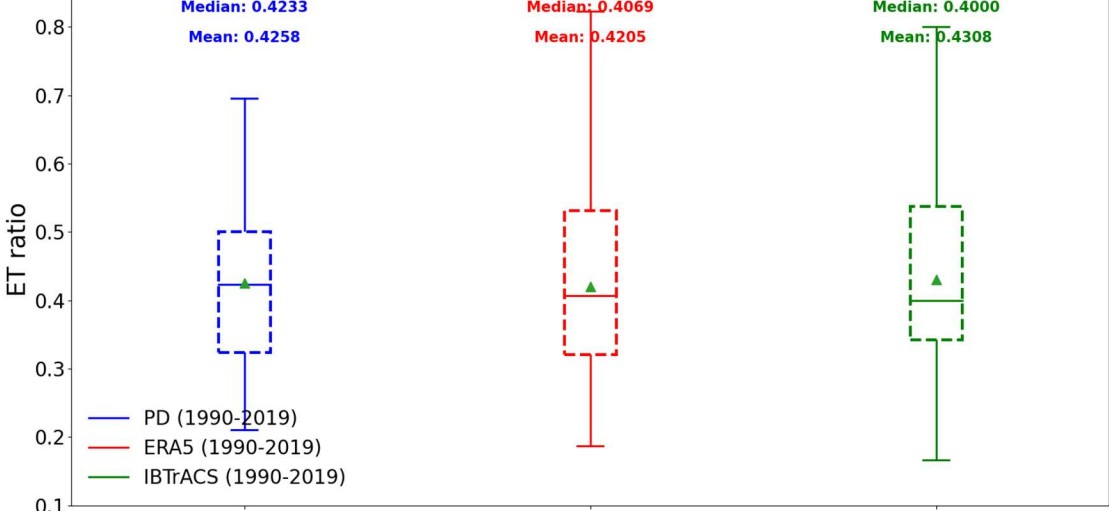


**Figure 1:** Box plot of the ET ratio for the present-day simulation (blue), ERA5 (red), and IBTRACS (green).
The box represents the interquartile range (IQR) containing 50% of the data; the upper edge of the box
represents the 75th percentile (upper quartile -UQ) while the lower edge is the 25th percentile (lower quartile
– LQ). The horizontal line within the box indicates the median, while the green triangle indicates the mean.
The whiskers extend to the smallest and largest data points within 1.5 times the IQR from the quartiles. Points
beyond the whiskers are considered outliers.


    **3.    RESULTS**

**3.1    Tropical cyclones in present-day simulations and future climate simulations**
The annual average number of TCs, including tropical storms, is significantly lower (-3.7) in the future
climate simulation (14.3) than in the present-day simulation (18). The season's peak remains in September
for the future climate simulation (Fig. 2). These results are consistent with other studies indicating an overall
decrease in TC frequency (Bender et al., 2010; Knutson et al., 2020; Mallard et al., 2013)

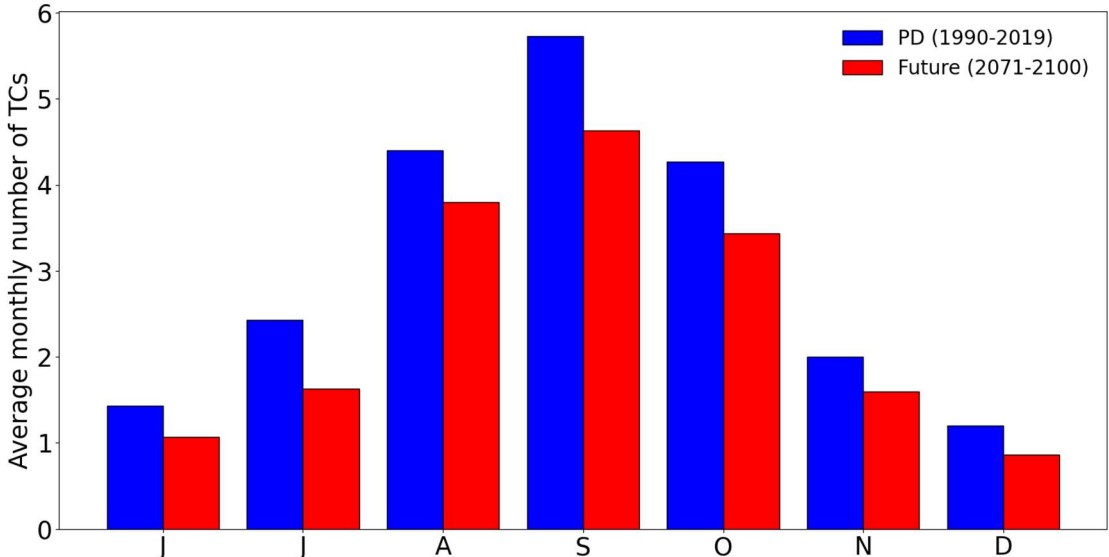


**Figure 2:** Average number of monthly TCs for the present-day (blue) and the future climate simulation (red)
from June to December
For each tracked TC, the maximum intensity – defined here as the minimum pressure reached by the cyclone
along its trajectory – was determined. Therefore, the latitude of the minimum pressure is not inherently
dependent on ET: it can occur either before or after transition.
Consistent with previous studies (Hill & Lackmann, 2011; Knutson et al., 2020; Kossin et al., 2020), we
found that there are more extreme events in the future climate simulation than in the present-day simulation
(Fig. 3) and that the mean storm minimum pressure is deeper in the future climate simulation (-3 hPa).
Consistent with other studies (Lee et al., 2020; Studholme et al., 2022), the median TC maximum intensity
is slightly shifted northwards of about 1.2° latitude (Fig. 4 a) in a warmer climate because of higher SST that
help TCs sustain their intensity at higher latitudes.

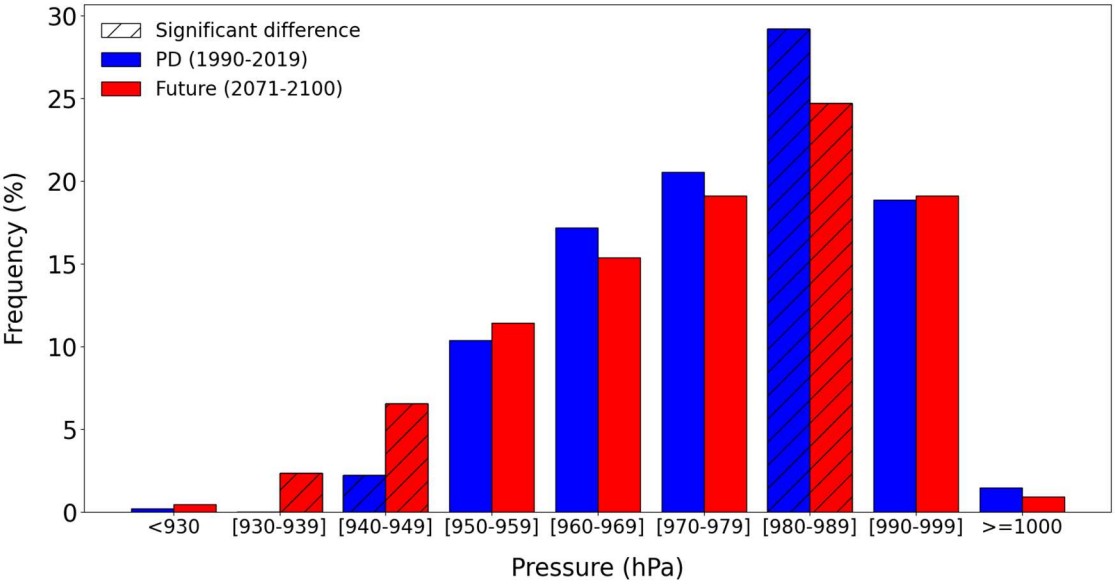

**Figure 3**: Distribution of the maximum intensity for the present-day (blue) and the future climate simulations (red). The maximum intensity is defined as the minimum pressure reached at the TC center during its lifetime. The hatched bars correspond to the intensity ranges with a significant difference.

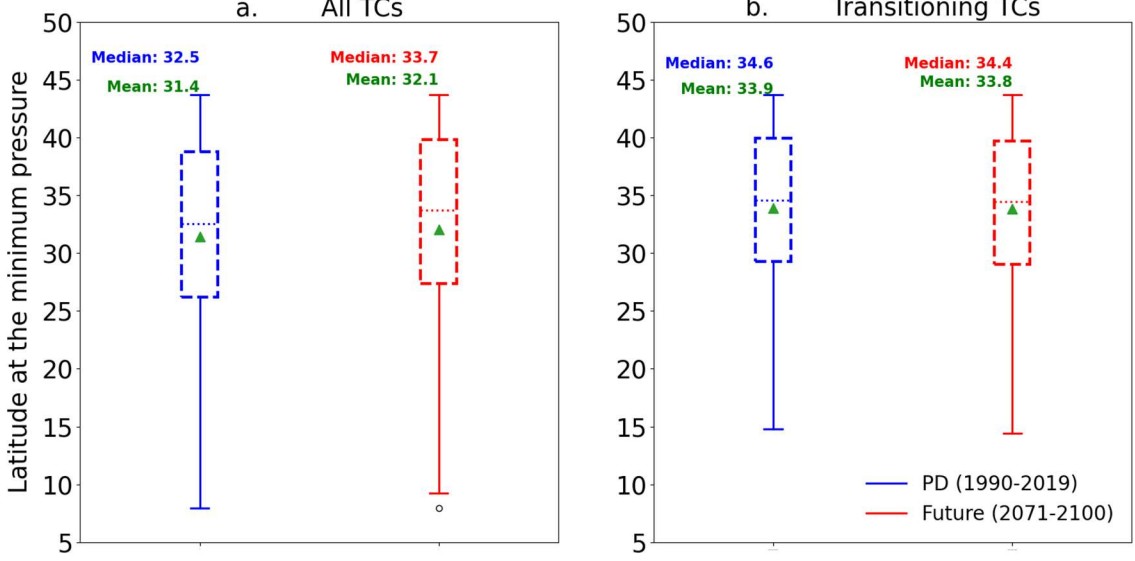

**Figure 4**: Box plot of the latitude of the minimum pressure for the present-day (blue) and the future climate (red) simulations a) for all TCs and b) for transitioning TCs. The box represents the interquartile range (IQR) containing 50% of the data; the upper edge of the box represents the 75[th] percentile (upper quartile- UQ) while the lower edge is the 25[th] percentile (lower quartile - LQ). The horizontal line within the box indicates the median, while the green triangle indicates the mean. The whiskers extend to the smallest and largest data points within 1.5 times the IQR from the quartiles. Points beyond the whiskers are considered outliers.

**3.2 Change in the atmospheric baroclinicity**
As expected, the Eady Growth Rate in the future climate simulation is weaker than in the present-day
simulation (Fig. 5a). This can be attributed to Arctic polar amplification which reduces the thermal gradient
between the high and tropical latitudes, resulting in a weaker baroclinic zone (Barnes & Polvani, 2015;
Francis & Vavrus, 2012; Serreze et al., 2009). The difference is particularly pronounced at the mid-latitudes
and on the western side of the North Atlantic Ocean, where most transitions usually occur.
Conversely, the upper-tropospheric Eady Growth Rate is slightly higher in the future climate simulation than
the present-day simulation (Fig. 5b). That is consistent with the tropical upper-troposphere warming effect,
which increases the upper-tropospheric thermal gradient (Barnes & Polvani, 2015; Harvey et al., 2014;
Lorenz & DeWeaver, 2007).

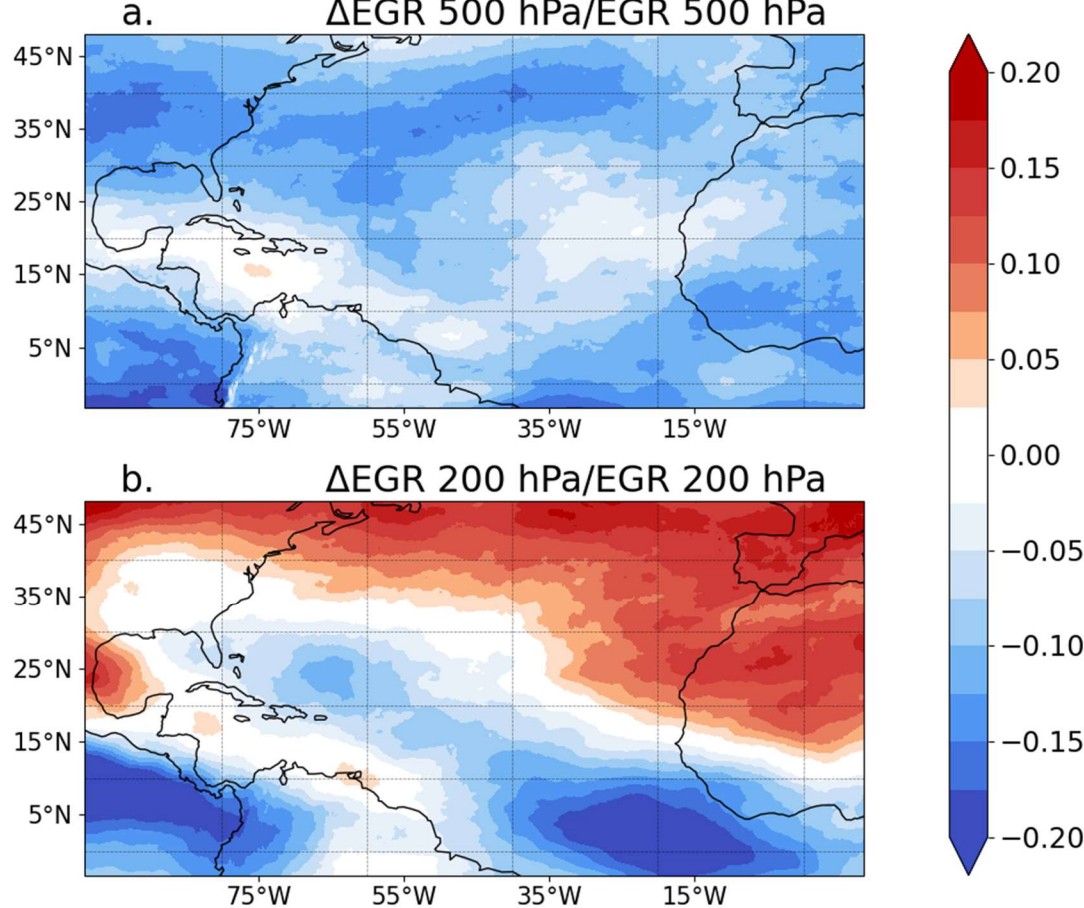


**Figure 5**: a) Relative difference in 500-hPa Eady Growth Rate between the future climate and the present-
day simulations and b) Relative difference in 200-hPa Eady Growth Rate between the future climate and the
present-day simulations
**3.3 ET events and ET ratios**
The mean annual number of ET events simulated in the future climate simulation (6.1) is significantly lower
than in the present-day simulation (7.6). The ET ratio, defined as the ratio of the number of ET events to the
number of TCs, is almost identical in the future climate simulation (42.7%) than in the present-day
simulations (42.6%). Our results are consistent with Bieli et al. (2020) which did not reveal any statistically
significant change in the ET rate in the North Atlantic. However, our findings contrast with the studies by
Liu et al. (2017)) and Baker et al. (2022), which reported a slight increase in ET frequency in the North
Atlantic basin.
Hart & Evans (2001) highlighted that ET events are more likely to occur if TCs maintain a minimum level
of intensity when they encounter a relatively strong baroclinic zone, enabling them to release the available
potential energy of the atmosphere. This minimum intensity level generally corresponds to a theoretical
minimum pressure of 960 hPa (Bister & Emanuel, 1998; Hart & Evans, 2001). Our findings indicate that the
northward shift of the baroclinic zone (Fig. 6a) is balanced by a corresponding northward shift in the 960-
hPa theoretical minimum pressure (Fig. 6b), thereby maintaining the relative position of this minimum
intensity level with respect to the favorable baroclinic regions. As a result, these factors may help to partially
explain why the probabilities of ET do not show significant differences.

| Simulation | Mean annual number of ET events | Mean ET ratio |
|---|---|---|
| **Present-Day** | 7.6 | 42.6% |
| **Future Climate** | 6.1 | 42.7% |


**Table 2**: Mean annual ET numbers and mean annual ET ratio for present-day and future climate simulations.

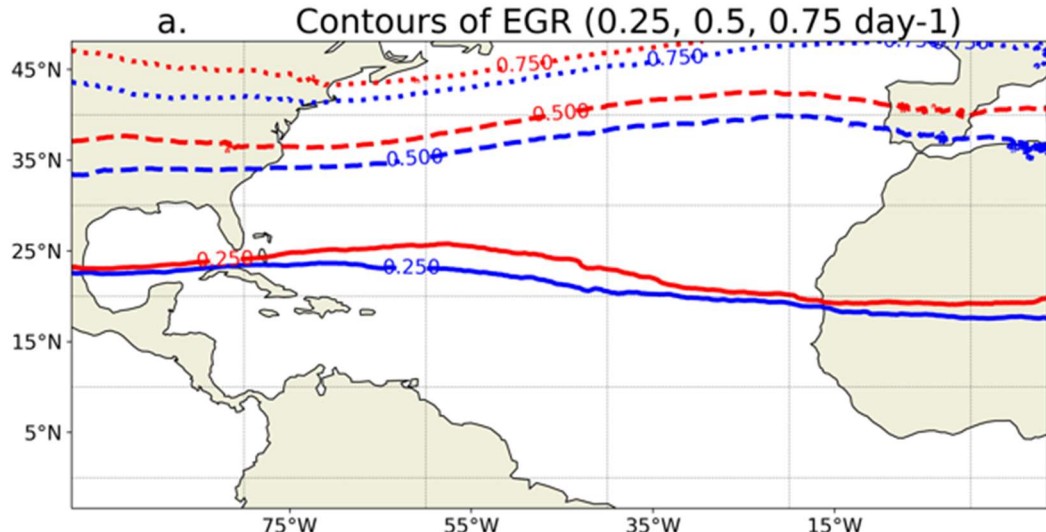

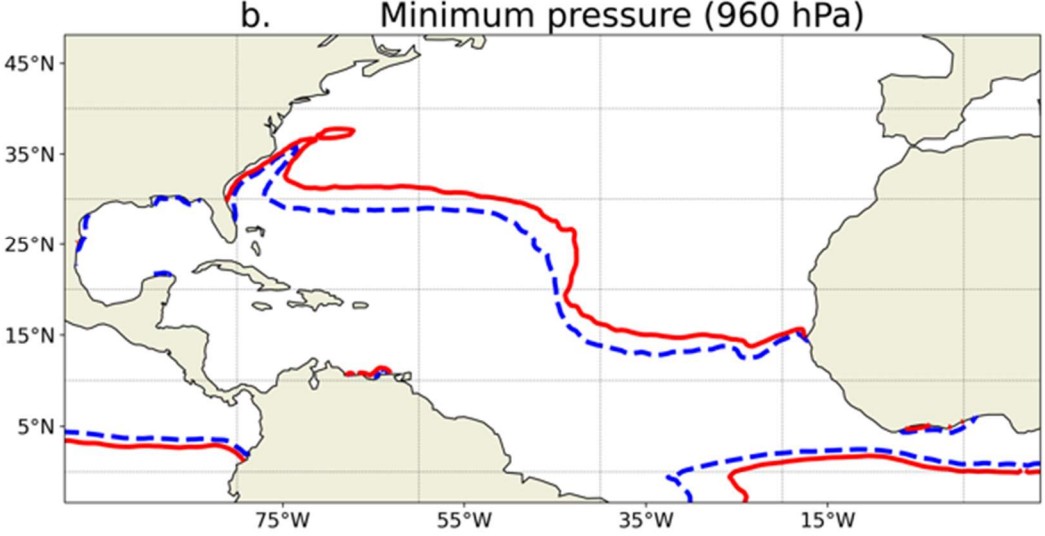

**Figure 6**: (a) Contours of mean Eady Growth rate for the present-day (blue) and the future climate (red) simulations. The solid lines represent the 0.25 day$^{-1}$ level, the dashed lines represent the 0.5 day$^{-1}$ level, and the dotted lines represent the 0.75 day$^{-1}$ level. (b) Contours of the 960-hPa theoretical pressure for the present-day (blue dashed line) and the future climate (red solid line) simulations

### 3.4    ET seasonal cycle in future climate

To assess changes in the seasonality of ET events, the mean annual contribution of each month to the mean annual ET ratio was calculated. This indicator is calculated as follows: for each year, the ET ratio is the number of ET events divided by the total number of TCs and then averaged over 30 years. This approach highlights the months when ET is most likely to occur, accounting for both the probability of TC occurrence and the conditional probability of ET.

In the present-day simulation, September and October are the most significant contributors to the annual ET ratio, as highlighted in Hart & Evans (2001). During these two months, the number of TCs and the baroclinic

energy remain relatively high, providing favorable conditions for ET events. In the future climate simulation,
September and October remain the months when the baroclinic energy levels are highest. However, the ET
season's peak appears to have shifted from September to October (Fig. 7). Indeed, in the future climate
experiment, the simulated decrease in October ET events mean number is less pronounced than the simulated
decrease in October mean number of TCs, suggesting a greater ET probability.

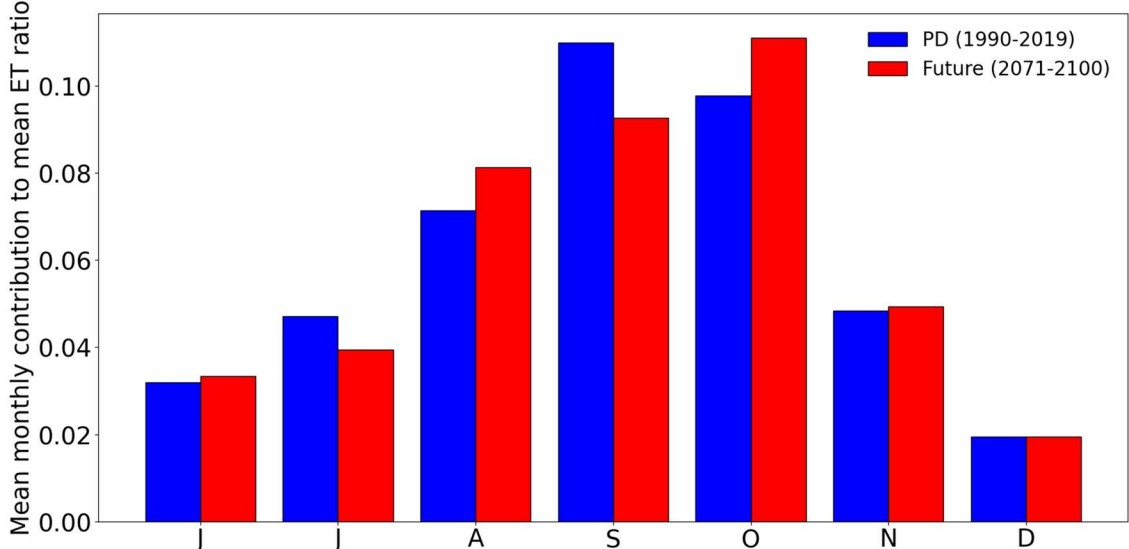

**Figure 7**: Monthly contribution to mean annual ET ratio for the present-day (blue) and the future climate
(red) simulations from June to December

### 3.5    Location of ET onsets
In this section, we focus on the impacts of ET locations, particularly to assess the threats they may pose to
the U.S. and Canada coastal populations.
In both experiments, TCs that undergo ET reach their maximum intensity at higher latitudes compared to
those that do not undergo ET (Fig. 4 a and b). Indeed, TCs that are most likely to undergo ET need to sustain
a minimum energy level at middle latitudes (Hart & Evans, 2001). However, no significant northward shift
in maximum intensity location for TC undergoing ET is simulated (Fig. 4b). This finding partly explains
why, despite the previously highlighted northward shift in the baroclinic zone in future climate simulations
(Fig. 6a), transitions do not occur further north (Fig. 8a) in the future climate simulation. These observations
indicate that the mean meridional displacement between the maximum intensity and the ET onset locations
does not significantly change under climate change. Additionally, the results show no significant change in
the mean longitude of ET onsets (Fig. 8b). Our results slightly contrast with Bieli et al. (2020) that show a
equatorward migration of the ET onset latitude, this shift being small in the North Atlantic basin. The
differences in the ET tracking methodologies might explain this difference.

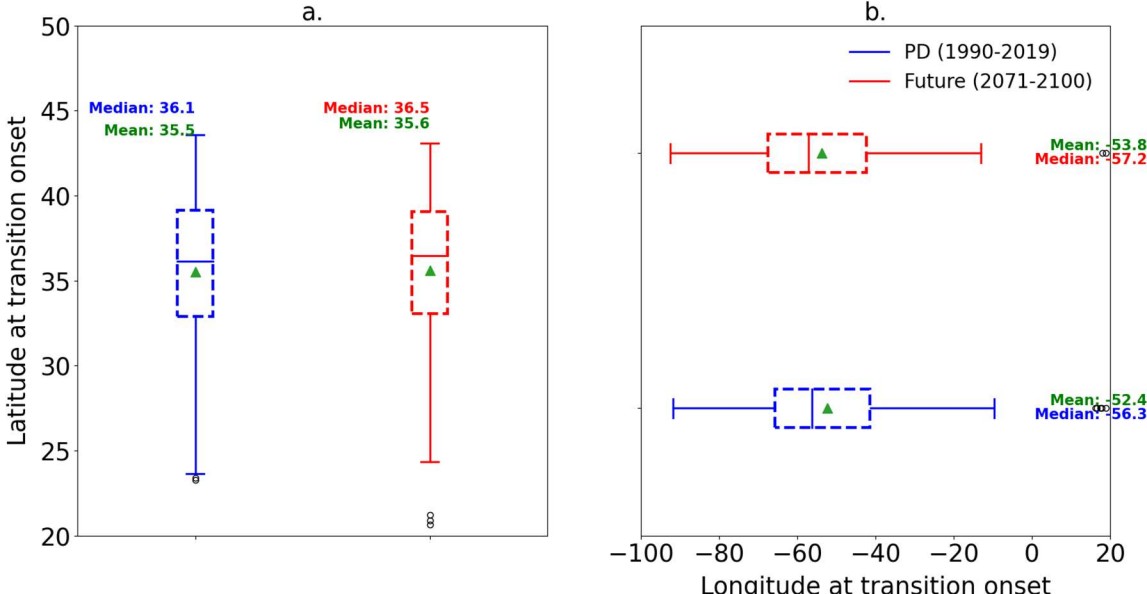

**Figure 8**: Box plot of the a) latitude of ET onset for the present-day (blue) and the future climate (red) simulations and b) longitude of ET onset for the present-day (blue) and the future climate simulations (red). The box represents the interquartile range (IQR) containing 50% of the data; the upper edge of the box represents the 75[th] percentile (upper quartile - UQ) while the lower edge is the 25[th] percentile (lower quartile - LQ). The horizontal line within the box indicates the median, while the green triangle indicates the mean. The whiskers extend to the smallest and largest data points within 1.5 times the IQR from the quartiles. Points beyond the whiskers are considered outliers.

The density map of ET onset, estimated with a Gaussian kernel, shows some differences (Fig. 9), with more ET onsets occurring near the U.S. Northeastern coast around 35°N and 40 °N. This region corresponds to the zone where a pronounced northward shift in the theoretical minimum pressure is simulated in the future climate simulation compared to the present-day experiment (Fig. 6b). This result does not agree with previous studies (Baker et al., 2022; Bieli et al., 2020; Liu et al., 2017) which reported more storms undergoing ET in the central and eastern North Atlantic, which leads to more storms, with a tropical origin, hitting Western Europe (Baatsen et al., 2015; Haarsma et al., 2013).

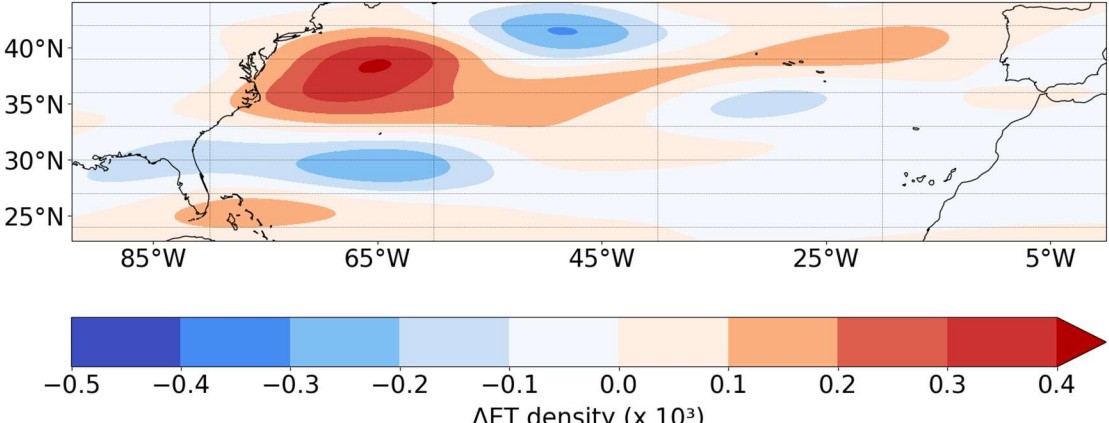

**Figure 9**: Difference in onset ET density between the future climate and the present-day simulations.

The overall lack of change in the mean ET onset latitude in the future climate simulation might be explained by stronger tropical cyclones, which have slightly deeper low pressure (982 hPa compared to 986 hPa today) at the time of ET onset, compensating for the weaker mid-tropospheric baroclinic zone that drives energy release. Additionally, the upper-tropospheric baroclinic zone becomes stronger, further offsetting the mid-tropospheric weakening. As a result, these factors balance out, preventing significant shifts in the average latitude of extratropical transition onset.

### 3.6 Duration of ET in Future Climate

Here we investigate a potential change in the duration of ET events as high SSTs have been associated with slow-transitioning storms, which are generally stronger than fast-transitioning storms (Hart et al. ,2006). The ET duration is defined as the time difference between the ET onset and ET completion. For ET events that are not completed within the regional domain, the ET completion time is defined as the time when the storm reaches the upper boundary of the spatial area. The performance of the CPS methodology for calculating the ET duration was discussed by Kofron et al. (2010).

The analysis shows no significant change in the ET duration for the future climate simulation compared to the present-day experiment (Fig. 10 a). This conclusion also holds for storms where the transition is completed within the regional domain (Fig. 10 b).This result contrasts with the findings of Jung & Lackmann (2019) which revealed an extended ET period. However, this conclusion applies only to a specific storm, and the characteristics of its track may influence the results. Our findings are, nevertheless, consistent with the results of Michaelis & Lackmann(2021) who found no statistically significant difference in the ET duration time between present-day and future climate simulations.

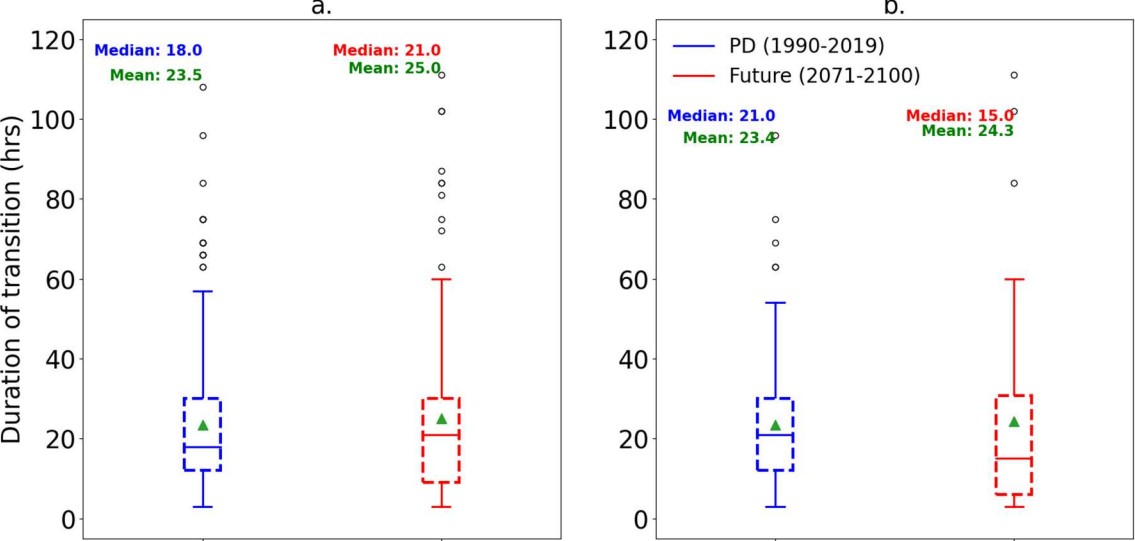

**Figure 10**: Box plot of the transition duration (in hours) for the present-day experiment (blue) and the future climate simulations (red) for: a) all storms, and b) storms for which the transition is completed within the regional zone. The box represents the interquartile range (IQR) containing 50% of the data; the upper edge of the box is the 75th percentile (upper quartile - UQ) while the lower edge is the 25th percentile (lower quartile - LQ). The horizontal line within the box indicates the median, while the green triangle indicates the mean. The whiskers extend to the smallest and largest data points within 1.5 times the IQR from the quartiles. Points beyond the whiskers are considered outliers.

**3.7    Energetics of Transitioning Storms in Future Climate**
This section explores the energetic changes in transitioning storms under future climate scenarios, focusing
on how their destructive potential evolves and the factors contributing to these changes. The destructive
potential of transitioning storms is notably higher (+20.5%) in the future climate simulation relative to present
day. This increase is reflected in the cumulative IKE over the transition period, which is significantly higher
in future climate simulations (Fig. 11). This increased destructive potential is partly attributed to a
significantly higher latent heat flux (+17%, Fig. 12 a) in the future climate simulation during the transition,
driven by higher SSTs. As expected, the Eady Growth Rate is significantly weaker in the future climate
simulation (Fig. 12 b), suggesting a reduction in baroclinic conversion.
Our findings align with previous studies (Cheung & Chu, 2023; Jung & Lackmann, 2021, 2023) which
highlighted the increase in storm intensity during ET.

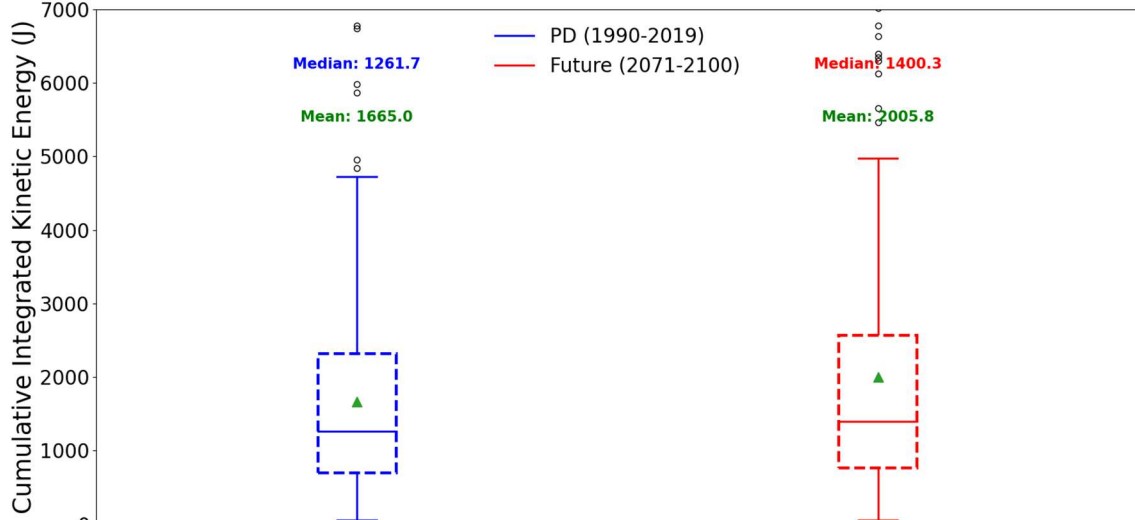


**Figure 11**: Box plot of Cumulative Integrated Kinetic Energy (in Joules) during the transition for the present-
day experiment (left) and the future climate simulation (right). The box represents the interquartile range
(IQR) containing 50% of the data; the upper edge of the box represents the 75th percentile (upper quartile -
UQ) while the lower edge is the 25th percentile (lower quartile - LQ). The horizontal line within the box
indicates the median, while the green triangle indicates the mean. The whiskers extend to the smallest and
largest data points within 1.5 times the IQR from the quartiles. Points beyond the whiskers are considered
outliers.

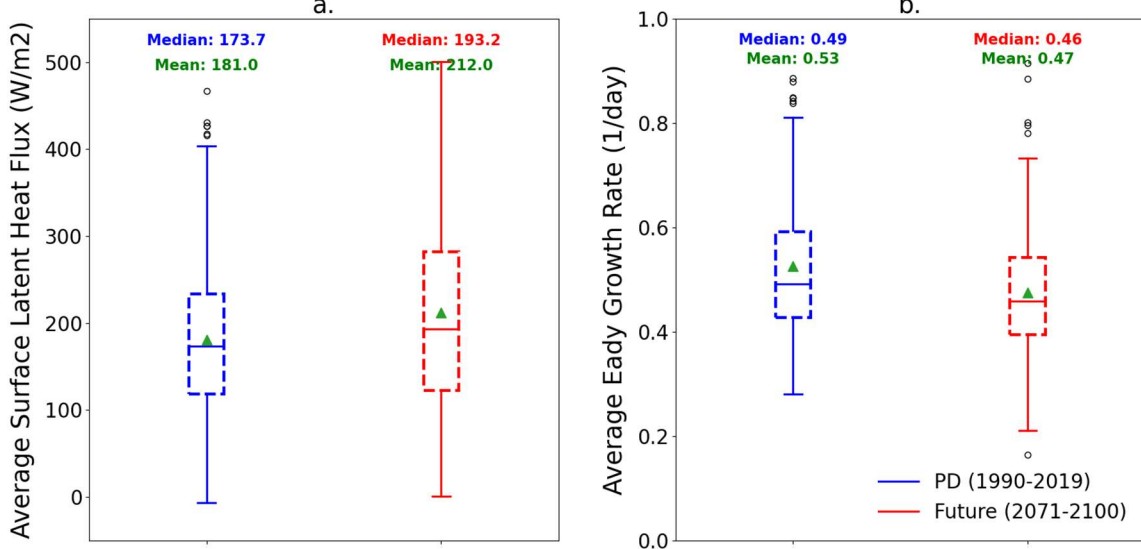


**Figure 12**: Box plot for the present-day experiment (blue) and the future climate simulation (red) for: a) the Surface Latent Heat Flux during, and b) the average 500-hPa Eady Growth Rate during the transition. The box represents the interquartile range (IQR) and contains 50% of the data; the upper edge of the box represents the 75th percentile (upper quartile - UQ) while the lower edge is the 25th percentile (lower quartile - LQ). The horizontal line within the box indicates the median, while the green triangle indicates the mean. The whiskers extend to the smallest and largest data points within 1.5 times the IQR from the quartiles. Points beyond the whiskers are considered outliers.

### 3.8    Reintensification during transition

Reintensification of storms during the ET phase is a critical aspect to evaluate as it influences the overall impact and longevity of transitioning storms. Reintensification during the transition phase is assessed using pressure differences and changes in IKE. The analysis reveals that storms, on average, do not intensify during the transition, with no significant difference in the pressure change (Fig. 13 a). On average, there is a slight increase in pressure for both experiments: +3.5 hPa for the present-day simulation and +4.5 hPa for the future climate simulation. Additionally, the relative difference in IKE also shows no significant variation between the present-day and the future climate simulations (Fig. 13 b). Despite the increase in storm central pressure, there is a modest rise in IKE during the transition for both climate states (+6.6% for the present-day experiment and +7.5% for the future climate simulation), potentially driven by the increase in storm size during the transition (Kozar & Misra, 2014).

The enhanced latent heat release has also been showed by Jung & Lackmann (2019) in a case study of the projection of Hurricane Irene (2011) in a warming environment. This has been interpreted as the primary cause of the intensification of the future transitioning storm Irene. Jung & Lackmann (2023) also highlighted a lesser risk of reintensification linked to a reduced baroclinic conversion. However, they focused on the post-ET intensification while our study investigates the intensification during ET.

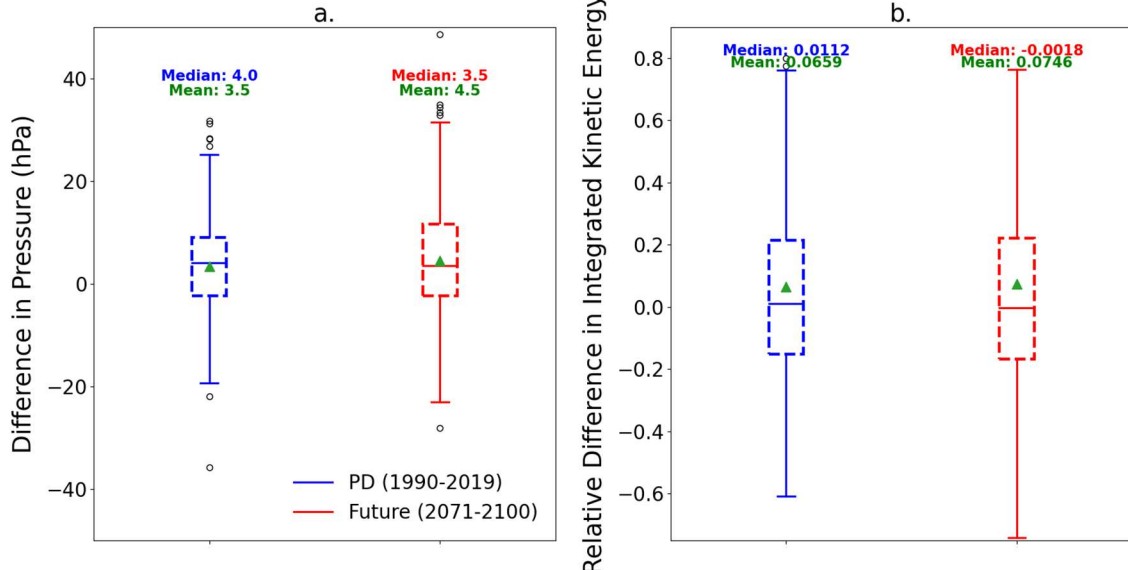

**Figure 13**: a) Box plot in difference in pressure at the storm center during the transition for the present-day (blue) and the future climate (red) simulations and b) Box plot in relative difference in Integrated Kinetic Energy (for present-day simulations (blue) and future climate simulations (red) during the transition. The box represents the interquartile range (IQR) containing 50% of the data; the upper edge of the box represents the 75th percentile (upper quartile - UQ) while the lower edge is the 25th percentile (lower quartile - LQ). The horizontal line within the box indicates the median, while the green triangle indicates the mean. The whiskers extend to the smallest and largest data points within 1.5 times the IQR from the quartiles. Points beyond the whiskers are considered outliers.

## 4. DISCUSSION AND CONCLUSIONS

This study investigates how extratropical transitions (ETs) in the North Atlantic basin might change by the end of the century under the RCP 8.5 climate scenario, using high-resolution climate simulations. While we found no significant difference in ET frequency, with the ET ratio (42.7%) in the future climate simulation being nearly identical to that in the present-day simulation (42.6%), our results indicate that transitioning storms in the future have greater potential destructiveness. Specifically, the Integrated Kinetic Energy associated with transitioning storms is significantly higher in the future climate simulation, driven largely by increased surface latent flux rather than enhanced baroclinic energy. This result aligns with the findings by Cheung and Chu (2023), which also reported an increase in the potential destructiveness of ETs.

While our findings about the ET frequency results contrast with studies by Liu et al. (2017) and Baker et al. (2022), which reported a slight increase in ET frequency in the North Atlantic basin, our results are consistent with Bieli et al. (2020) which did not report any significant change in the ET frequency and with previous research indicating that TCs will become less frequent but more intense in the future (Bender et al., 2010; Knutson et al., 2020; Mallard et al., 2013). Our simulations also confirm a poleward migration of the maximum intensity of TCs (Lee et al., 2020), aligning with the expansion of TC cyclogenesis regions.

The findings indicate that the future climate simulation shows a weakening and northward shift in the mid-tropospheric baroclinic zone, driven by polar amplification, along with a slight increase in the upper-tropospheric baroclinic zone due to warming in the tropical upper troposphere. This weakening of the baroclinic zone along with the decrease in the number of TCs explain the decrease in the number of ET events, which ultimately leads to the stability of the ET frequency.

Additionally, our results do not show a significant change in ET seasonality, with September and October
remaining the primary months for ET events. However, the peak's season seems to have shifted from
September to October, suggesting that large-scale environmental conditions may become more favorable for
ET in October in the future climate simulation.
No significant shift in the latitude of ET onsets is simulated in the future climate simulation, although there
is a slight increase in ET occurrences near the U.S. Northeastern coast. This could be due to more intense
TCs reaching favorable baroclinic zones, which contrasts with Baker et al. (2022) who reported a decrease
in ET occurrences in this region, mainly explained by the poleward and eastward expansion of the
cyclogenesis region. Our findings also contrast with Bieli et al. (2020) who reported a slight equatorward
shift.
Previous studies (Jung & Lackmann, 2019) suggested that the duration of ETs might be longer in the future
due to higher SSTs, an empirical indicator of slow-transitioning storms (Hart et al., 2006), and due to reduced
meridional SST gradient, which inhibits baroclinic conversion. However, despite an environment that is less
baroclinic during ET, no significant difference in duration is simulated as shown in Michaelis & Lackmann,
(2021). There could be some biases in our findings as some storms have not completed their transition within
the spatial domain, even though no significant difference in ET has been found for storms that have completed
their transition. Indeed, Kitabatake (2011) and Kofron et al. (2010) highlighted the limitations of the Cyclone
Phase Space in the detection of the ET duration. In addition to that, the inability of the Cyclone Phase Space
to resolve the cyclone's inner-core structure (Evans et al., 2017) may contribute to this finding.
Within the spatial zone considered, our study suggests that transitioning storms do not necessarily reintensify
more in a warming environment, consistent with the findings of Jung and Lackmann (2023), due to a
reduction in baroclinic conversion. However, our study mainly focused on the reintensification during
transition. Further work would be needed to investigate the potential reintensification post ET.
In conclusion, our study suggests that extratropical transitions will pose a greater risk for populations in the
U.S. Northeastern coast and the Maritimes. However, uncertainties remain regarding the impact of global
warming on ET frequencies and the spatial and temporal distribution of ET events. Further research is needed
to address these uncertainties.
Future studies should investigate the large-scale environmental conditions affecting the Northern
Hemisphere, including East Pacific and North America. Hart and Evans (2006) emphasized that storms are
more likely to intensify after interacting with a negatively-tilted rather than a positively-tilted trough.
Therefore, a better understanding of how climate change will impact the occurrence of negative-tilted versus
positive-tilted troughs will be crucial for grasping future ET dynamics.
Additionally, the structure of post-transition storms warrants further exploration. Hart and Evans (2006)
noted that warm-seclusion cyclones, which are more likely to cause damage, should be examined in the
context of global warming. Assessing how global warming affects the post-transition structures of storms
will enhance our understanding of future risks associated with ETs.
Hart and Evans (2001) also mentioned that 50% of tropical cyclones making landfall between 1950 and 1996
were transitioning storms. Investigating the impact of global warming on the spatial pattern of transitioning
storms that make landfall will be important for anticipating future damages.
Finally, the simulations used in our study were atmospheric-only experiments with prescribed SST. Baker
et al. (2022) demonstrated that high-resolution fully-coupled simulations may yield different outcomes
compared to atmospheric-only simulations. For instance, while atmospheric-only simulations showed an
equatorward shift in the completion latitude, fully coupled simulations detected a poleward shift. Previous
studies have highlighted the necessity of considering the ocean's negative feedback mechanism on tropical
cyclones (Schade and Emanuel 1999; K. Emanuel et al. 2004). The winds associated with tropical cyclones
induce upwelling of cold waters, which cools the sea surface temperature and inhibits the intensification of
tropical cyclones (Schade and Emanuel 1999; K. Emanuel et al. 2004). The overestimation of the maximum
wind can reach up to 25 m·s⁻¹ (K. Emanuel et al. 2004). Scoccimarro et al. (2017) demonstrated that a high
coupling frequency could significantly reduce this bias. In the context of climate change, Huang et al. (2015)
showed that this ocean feedback is expected to strengthen due to the increased ocean stratification, which
could enhance the ocean's negative effect and reduce the expected intensification of tropical cyclones in
certain regions of the North Atlantic. Therefore, further investigations using fully coupled models are needed
to reconcile these discrepancies and build a comprehensive understanding of the impacts of climate change
on extratropical transitions.

## 5.    APPENDIX A: COMPARISON WITH IBTRACS

In this appendix, we further compare the model simulations with IBTrACS. Our results indicate an
overestimation (+2.2) of the average yearly number of TCs compared to observations (Fig. A1). However,
the distribution of storms per month is only slightly affected, with nonetheless an underestimation in
September and an overestimation in November and December (Fig. A2).

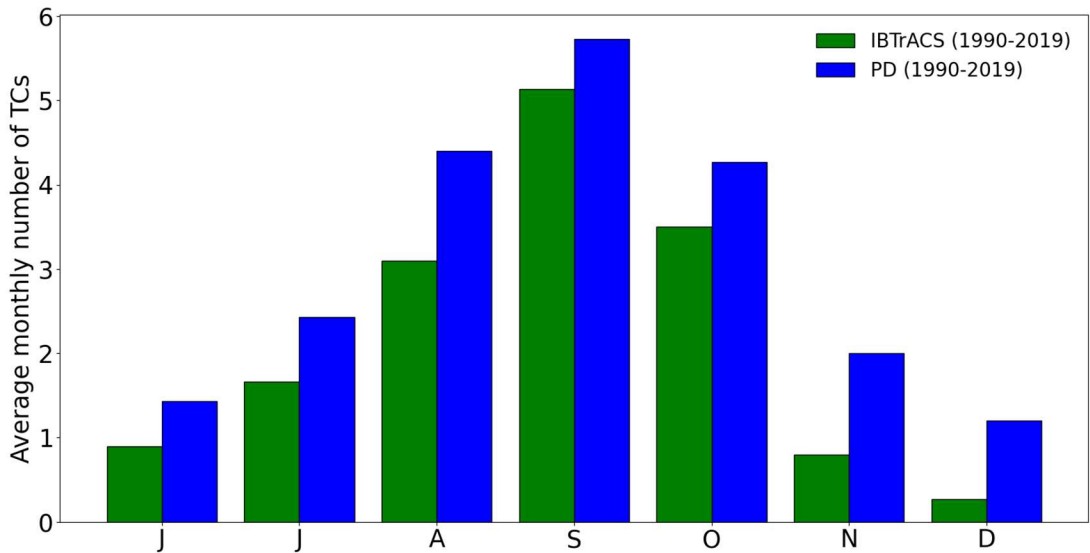

**Figure A1**: Average monthly number of TCs for IBTrACS (green) and present-day simulation (blue)

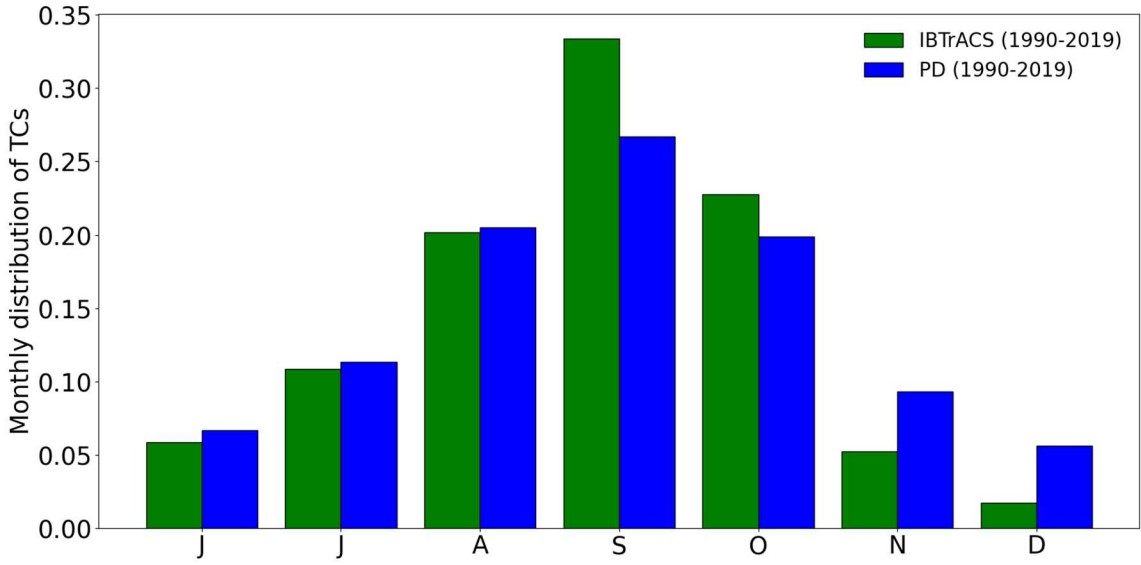


**Figure A2**: Monthly distribution of TCs for IBTrACS (green) and present-day simulation (blue)

Our results highlight challenges in accurately reproducing the most intense cyclones (CAT4 and CAT5). Tropical storms are also underestimated while CAT1, CAT2 & CAT3 categories tend to be overestimated (Fig. A3). The storm categories are based on the minimum pressure as detailed in Table A1.

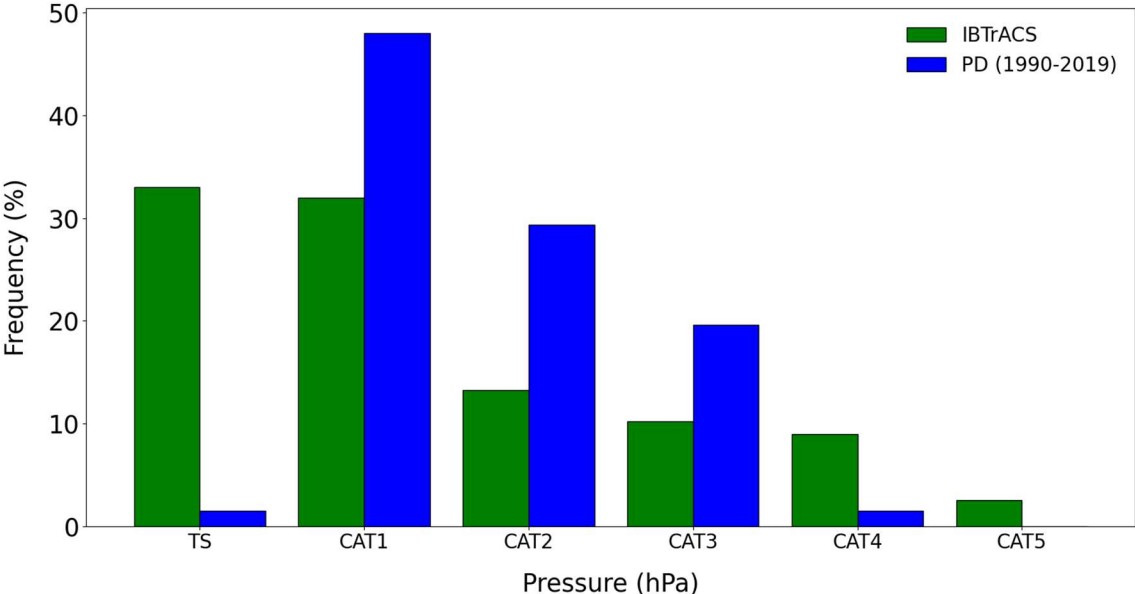


**Figure A3:** Distribution of tropical storms per category for IBTrACS (green) and present-day simulation (blue).





| Category | Minimum pressure (hPa) |
|:---:|:---:|
| TS | >1000 |
| CAT1 | [980 - 1000[ |
| CAT2 | [965 - 980[ |
| CAT3 | [945 - 965[ |
| CAT4 | [920 - 945[ |
| CAT5 | <920 |


**Table A1**: Storm category definition per minimum pressure range
A northward shift in the minimum pressure latitude is also present. Indeed, the average latitude is 31.4° for
the present-day experiment and 27.4° for the observations (Fig. A4). This northward shift is mainly explained
by the overestimation of the CAT1/CAT2/CAT3 categories whose minimum pressure latitude are the highest
(Table A2).

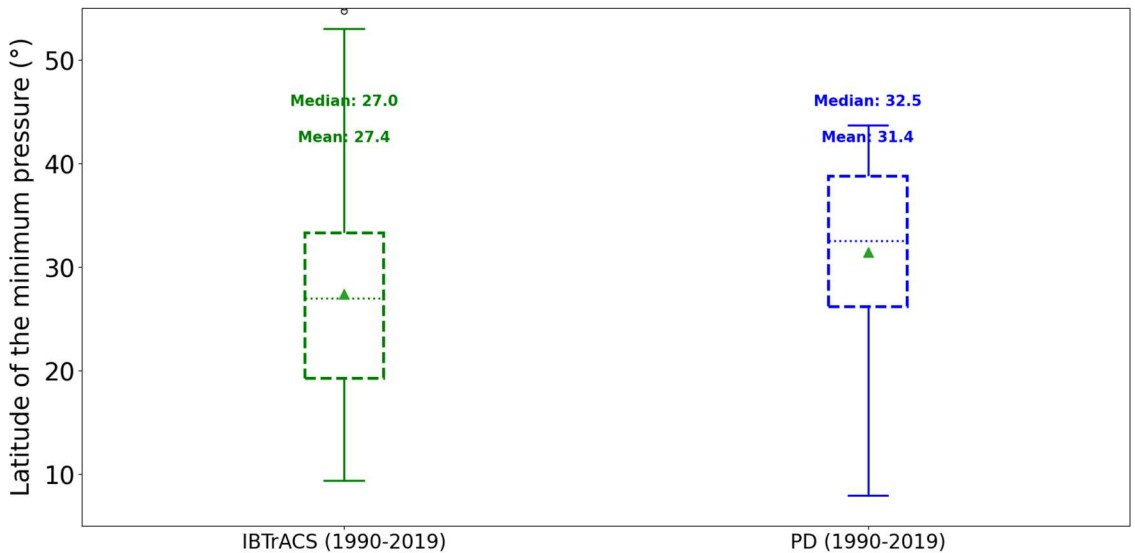


**Figure A4**:  Latitude of the minimum pressure for IBTrACS (green) and present-day experiment (blue).

| | Latitude of minimum pressure (°) |
|:---:|:---:|
| TS | 23.7 |
| CAT1 | 29 |
| CAT2 | 32.7 |
| CAT3 | 31.7 |
| CAT4 | 23.5 |
| CAT5 | 22.09 |


**Table A2:** Latitude of the minimum pressure for IBTrACS (1990-2019)
Overall, the model satisfactorily reproduces the TC density in the North Atlantic. However, some differences
should be noted (Fig. A5): the TC density is overestimated in the western North Atlantic (around the 55° W)
while it is underestimated near the U.S. East coast and the Gulf of Mexico coast. Additionally, TC density is
underestimated in the central North Atlantic between 10°N and 15°N.

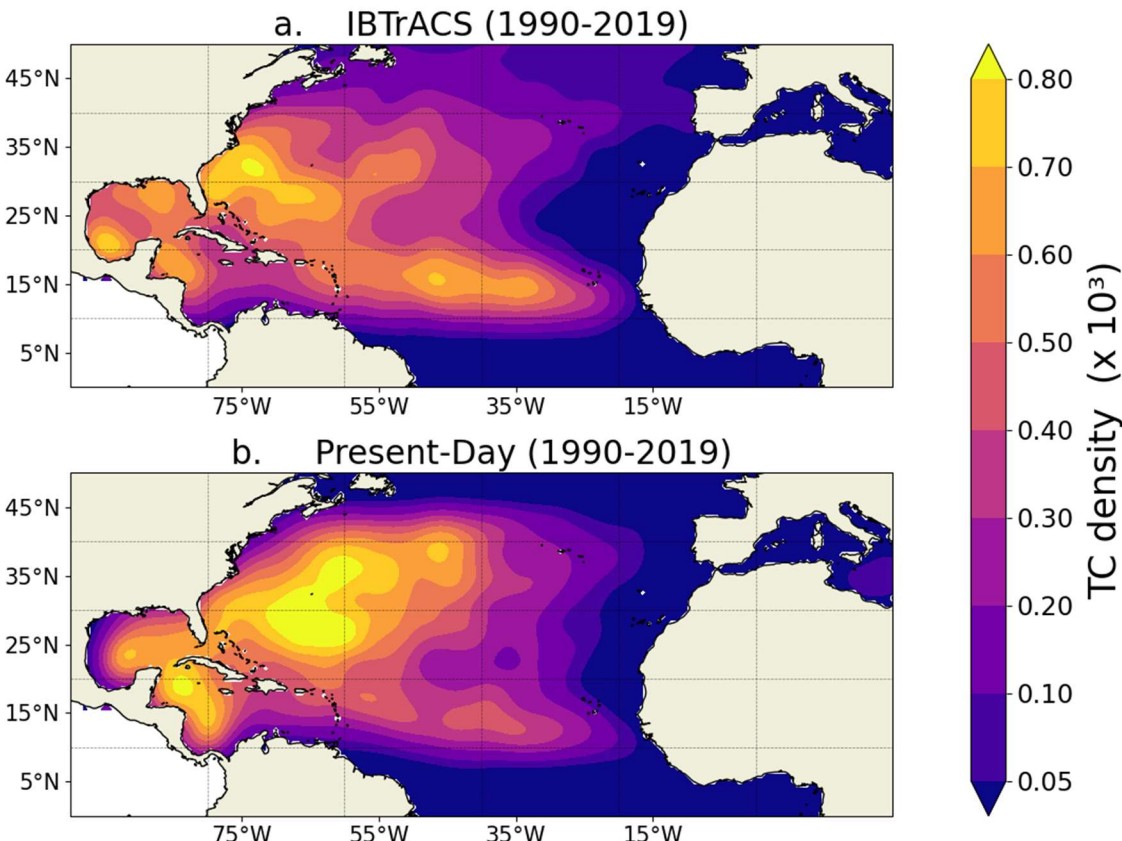


**Figure A5**:  TC density for IBTrACS and present-day experiment. The TC density maps are estimated using
a Gaussian Kernel.
**6.  CODE AVAILABILITY**
The codes used are available from the corresponding author upon request.
**7.  DATA AVAILABILITY**
The data used are available from the corresponding author upon request.
**8.  AUTHOR CONTRIBUTION**
**Aude Garin**: Methodology, Software, Validation, Formal Analysis, Investigation, Writing - Original Draft,
Visualisation. **Francesco S.R. Pausata**: Conceptualization, Methodology, Formal Analysis, Investigation,
Resources, Writing – Review & Editing, Visualisation, Supervision. **Mathieu Boudreault**:
Conceptualization, Formal Analysis, Investigation, Resources, Writing – Review & Editing, Visualisation,
Supervision. **Roberto Ingrosso**: Model Simulations, Software, Validation, Writing – Review & Editing.

## 9. COMPETING INTERESTS

The Authors declare that they have no conflict of interest.

## 10. ACKNOWLEDGEMENTS

The authors would like to thank Katja Winger for the help in developing the algorithms, Frédérik Toupin and Yassine Hammadi for the technical support.

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
