# Peer review of "The impacts of climate change on tropical-to-extratropical"

_EGUsphere, 2024_

## Author Comment (AC1)

**Authors' response to**
**'Comment on egusphere-2024-3435', Anonymous Referee #1, 10 Dec 2024**

We thank the referee for taking the time to review our manuscript. We believe the suggestions made by the referee greatly improved our manuscript. Below you will find the referee's comments in **bold**, our replies in blue, and *italics* for the text that has been modified/added to the manuscript. The additions are highlighted in turquoise in the revised manuscript.
* * *
**Garin et al. (2024) uses a regional climate model over two 30-year periods to examine the effects of climate change (under RCP8.5) on ET events in the North Atlantic. The authors find no significant change in the frequency of ET events in the future but a shift in their location (increase off the northeast coast) and increase in potential destructiveness.**

**Given the limited number of studies on ET and climate change, I appreciate this addition to the literature. The model simulations used in this study are high enough resolution to adequately capture TCs and ET events and storm tracking methods are in line with previous studies. I would, therefore, rate the scientific significance of this manuscript as "excellent-to-good".**

**The overall presentation quality is also "excellent-to-good" in the sense that the manuscript is concise and easy to follow. The scientific quality, however, is "good-to-fair" as substantial discussion of how the presented results compare with previous studies is omitted and should be included before publication. Additionally, I noted several omitted references for the authors to include in their introduction and/or to help put their findings into context.**

Thanks for your encouraging words and appreciation of our manuscript.

1. **Here are some additional references that should be included throughout the introduction and results. The Bieli et al. references are of particular interest to the current study:**

   o **Arnott et al. (2004): https://doi.org/10.1175/MWR2836.1**

   o **Baatsen et al. (2015): https://doi.org/10.1007/s00382-014-2329-8**

   o **Bieli et al. (2019): https://doi.org/10.1175/JCLI-D-17-0518.1**

   o **Bieli et al. (2020): https://doi.org/10.1029/2019MS001878**

   o **Haarsma et al. (2013): https://doi.org/10.1002/grl.50360**

   o **Kitabatake (2011): https://doi.org/10.2151/jmsj.2011-402**

   o **Kofron et al. (2010): https://doi.org/10.1175/2010MWR3180.1**

   o **Wood and Ritchie (2014): https://doi.org/10.1175/JCLI-D-13-00645.1**

   Thank you for your suggestions, we have added the aforementioned references in the introduction and results as suggested by the referee.

2.  **Of biggest concern is the lack of comparisons to previous studies throughout the results section. While there is some comparison on the Discussion and Conclusions section, the manuscript could benefit from additional comparisons and related discussion throughout. For each presented result, consider:**

    o   **How do these results compare to previous studies?**

    o   **What could account for the differences (e.g., methodologies, model environments, etc.)?**

We added comparisons with the literature throughout the manuscript as suggested by the referee. Please find below some *examples* we have added in the manuscript. The revised manuscript is provided in Track changes mode thus highlighting all changes made.

In Section 3.3, we have added the following sentences:

*"Our results are consistent with Bieli et al. (2020) which did not reveal any statistically significant change in the ET rate in the North Atlantic. However, our findings contrast with the studies by Liu et al. (2017) and Baker et al. (2022), which reported a slight increase in ET frequency in the North Atlantic basin."*

In Section 3.5, we have added the following sentences:

*"Our results slightly contrast with Bieli et al. (2020) that show a equatorward migration of the ET onset latitude, this shift being small in the North Atlantic basin. The differences in the ET tracking methodologies might explain this difference."*

In Section 3.6, we have added the following sentences:

*"This result contrasts with the findings of Jung & Lackmann (2019) which revealed an extended ET period. However, this conclusion applies only to a specific storm, and the characteristics of its track may influence the results. Our findings are, nevertheless, consistent with the results of Michaelis & Lackmann(2021) who found no statistically significant difference in the ET duration time between present-day and future climate simulations."*

**A couple small changes to the figures would be helpful to increase readability:**

o   **All box plot figures: Could be helpful for the reader to add grid lines and/or explicitly state the mean/median values either on the plots themselves or in the text.**

o   **Figure 1: Add legend on plot as in other figures.**

o   **Figure 3: Could be helpful to indicate which intensity ranges are significantly different in the future.**

o   **Figure 6: The information provided in this figure could be better suited for a table instead.**

We have explicitly stated the mean/median values on the plots themselves. Figure 6 has been replaced by Table 3 and a legend has been added to Figure 1. With respect to Figure 3, we have clarified that the mean pressure in the future is significantly deeper. For each intensity range, we have performed a statistical test between the present-day simulation data corresponding to this intensity range and the future climate simulation data corresponding to this intensity range. The intensity ranges with a significant difference have been hatched in the figure.

**Specific Comments**

**L93: The "ET" acronym was already defined in L28.**

> Thanks for having pointing out this, we have corrected it now.

**L106: In addition to precipitation validation, what data set was used to evaluate model TC tracks? I see some evaluation of the ET ratio in section 2.8 compared to IBTrACS and ERA5, but what about for the TC and ET tracks themselves? In particular, I would be curious to see how CRCM5/GEM 4.8 handles TCs in the eastern North Atlantic main development region.**

> The model TC tracks have been evaluated with IBTrACS. We have added the results of this evaluation in appendix A of the manuscript.

> With respect to ET tracks, we have added the table below and the following short discussion in Section 2.8:

> *"ET in IBTrACS is determined subjectively by various forecasters based on real-time observational data. In addition, IBTrACS' phase transition occurs at an instantaneous point in space and time and provides no information about the path of ET (Zarzycki et al., 2017). To assess the ability of the model to spatially reproduce ET, we have compared the latitude and the longitude of ET onset with the results of Bieli et al. (2019) in Table 2 [Table R2 below]. The comparison shows a northward shift in our simulated ET onset latitude compared to Bieli et al. (2019). This difference may be explained by our methodology, which in the case of multiple transitions, considers only the final transition. The eastward shift in the ET onset longitude is a consequence of the northward shift in the ET onset latitude, as storms tend to go eastward at higher latitudes."*

| Simulation | Mean Latitude ET Onset | Mean Longitude ET Onset |
|---|---|---|
| GEM 4.8/CRMC5 | 35.5 | -52.4 |
| JRA55 (Bieli et al., 2019) | 33.2 | -58.4 |
| ERA5 - Interim (Bieli et al., 2019) | 28.9 | -56.2 |

**Table R2**: ET onset mean latitude and longitude

**L108: Is the precipitation comparison shown anywhere in the manuscript? What does a reasonable precipitation comparison mean for the model's ability to represent the TC/ET climatology?**

> The precipitation comparison was a general evaluation of the model and indeed does not provide meaningful information in terms of the model's ability to represent TC/ET climatology. We have pointed this out in the manuscript when presenting the precipitation climatology. As stated previously, IBTrACS has been used to evaluate the ability of the model to represent the TC/ET climatology in Appendix A.

**L108: Is it possible to evaluate over the full 30-year simulation period? If not, please clarify and state this limitation.**

> The evaluation of the TC tracks has been evaluated over a full 30-year simulation period and we have added the results of this evaluation in Appendix A.

**L222: Remove extra space between "to" and "cold-core".**

> Thanks for having pointed this out; we have corrected it.

**L261: As noted in General Comment #2 above, it could be helpful to compare this model's simulated ET percentage to that from other modeling/observational studies.**

We have added a table of the mean annual ET ratio from other studies in the manuscript (Section 2.8)

**L274: Are 14.3 and 18 the annual averages? Please clarify.**

Yes, these numbers are annual averages. It now reads:

*"The annual average number of TCs, including tropical storms, is significantly lower (-3.7) in the future climate simulation (14.3) than in the present-day simulation (18)."*

**L280: Remove extra parenthesis after "studies".**

This was done.

**L299–301: Reference?**

We have added the following references:

Barnes & Polvani (2015)

Francis & Vavrus (2012)

Harvey et al. (2014)

Serreze et al. (2009)

**L304–305: Reference?**

We have added the following references:

Barnes & Polvani (2015)

Harvey et al. (2014)

Lorenz & DeWeaver (2007)

---

## Author Comment (AC2)

**Authors' response to**

**'Comment on egusphere-2024-3435', Anonymous Referee #2, 16 Dec 2024**

We thank the referee for taking the time to review our manuscript. We believe the suggestions made by the referee greatly improved our manuscript. Below you will find the referee's comments in **bold**, our replies in blue, and *italics* for the text that has been modified/added to the manuscript. The additions are highlighted in turquoise in the revised manuscript.
* * *
**The authors evaluate how extratropical transition (ET) in the North Atlantic will change with climate change using high-resolution regional climate simulations. They find a decrease in the number of ET events associated with a decrease in the number of tropical cyclones, with no change in the ratio of cyclone that undergo ET. They show that there is a compensation between more intensification from surface latent heat fluxes but a weakening baroclinic instability. They show a shift in the region of ET but no significant change in the average latitude. This is an excellent and well written paper about an interesting area of research. An excellent paper to have in this journal. I have some minor corrections and clarifications below.**

Thanks for your encouraging words and appreciation of our manuscript.

**L106 - Is there a missing reference here? Otherwise, you need to show this evaluation in a supplement.**

Yes, the reviewer is correct, we forgot to insert the reference: we have now added it. The model evaluation is available in Ingrosso and Pausata (2024).

**L186 - Is the weight of each layer the mass?**

The weight of each layer is calculated as the ratio of the difference between the upper-bound pressure and the lower-bound pressure of the layer to the difference between the upper-bound pressure and the lower-bound pressure of the entire column. We have included this definition of the weight in the manuscript.

**L203 - What do you mean by "i.e. upper" in this paragraph?**

We meant upper troposphere corresponding to the 600-300 hPa layer. We have clarified the definition of upper and lower troposphere in the manuscript. It now reads:

"*The lower troposphere corresponds to the 900-600 hPa layer while the upper troposphere corresponds to the 600-300 hPa layer*".

**L218 - What do you do with cyclones that are diagnosed as having an onset of ET but not completing ET? The paper by Sarro and Evans (2022) (https://doi.org/10.1175/MWR-D-22-0088.1) would be good to reference here. The "instant warm seclusion" they describe, where the cyclones undergo ET but are always warm core, could be relevant.**

Our study includes all TCs that have started an ET. Therefore, instant-warm seclusions, as well as transitioning storms that have not completed their transition within the regional domain are accounted for. Indeed, we have highlighted in the discussion the difficulties of

the Cyclone Phase Space methodology in identifying ET completion. We have now clarified this aspect in the methodology and add the reference you have suggested. It now reads:

*"Therefore, all TCs that have started an ET are included in our study, including instant-warm seclusions (Sarro & Evans, 2022), as well as transitioning storms that have not completed their transition within the regional domain."*

**L261 - I'm actually surprised at how close this is. I would have thought that IBTrACS underestimates ET due to reporting biases. Could you comment on this?**

Indeed, there can be reporting biases as mentioned by the reviewer: extratropical transitions in IBTrACS are determined subjectively by various forecasters based on real-time observational data (Zarzycki et al., 2017).

Several studies have explored the topic of ET ratio simulation in different basins over the past years using the CPS methodology with different models, resolutions or reanalyses. The simulated ET ratios that we have summarized in Table R1 and added to the manuscript, are highly diverse and are generally higher than the observations (Hart & Evans, 2001).

Bieli et al. (2019) used JRA-55 and ERA-Interim whereas Zarzycki et al. (2017) used two reanalysis products, ERA-Interim and CFSR, combined with two climate models, CAM-55 et CAM-28. The latter study highlights the importance of the resolution with a 9% increase in the mean annual ET ratio with a higher resolution. Liu et al. (2017) used two reanalysis products, CFRS and JRA-55, combined with two climate models, FLOR et FLOR-FA, for which the SST is artificially corrected through flux-adjustment. This correction leads to a better representation of the ET ratio. Studholme et al. (2015) found a very high mean annual ET ratio (68%), this finding being explained by the simulation of longer tracks, enabling the ET to occur.

The simulated ET ratios that we have summarized in Table R1 and added to the manuscript, are highly diverse and are generally higher than the observations (Hart & Evans, 2001).

However, we can point out that Bieli and al. (2019) or Zarzycki et al. (2017) have also simulated mean annual ET ratios which are close to the observations. In our paper, the ET ratio found in the present-day simulation is 42.7%. However, this value takes into account the adjustments made to the CPS method, as detailed in the Methodology section (L215-216). Indeed, we noticed that for certain tracks, some storms could begin to acquire extratropical characteristics (asymmetry or a cold core) before reverting to tropical cyclones. These "false" transitions were therefore excluded from the transitions. It is important to point out that if another transition occurs, the storm will be considered among the transitioning storms.

After accounting for these "false" transitions, the transition rate decreased from 68.5% (close to the findings of Studholme et al., 2015) to 42.7%.

| Author(s) | Mean ET fraction | Method/data for tracking ETs |
|---|---|---|
| Hart & Evans (2001) | 46% | NHC best track labels |
| Studholme et al. (2015) | 68% | CPS and k-means clustering, storms tracked in ECMWF operational analysis |
| Zarzycki et al. (2016) -1 | 55% | CPS, storms tracked in ERA-Interim |
| Zarzycki et al. (2016) - 2 | 50% | CPS, storms tracked in CFSR |
| Zarzycki et al. (2016) - 3 | 49% | CPS, storms tracked in CAM-28 |
| Zarzycki et al. (2016) - 4 | 40% | CPS, storms tracked in CAM-55 |
| Liu et al. (2017) - 1 | 56% | CPS, storms tracked in CFRS |
| Liu et al. (2017) - 2 | 50% | CPS, storms tracked in JRA-55 |
| Liu et al. (2017) - 3 | 57% | CPS, storms tracked in FLOR-FA |
| Liu et al. (2017) - 4 | 31% | CPS, storms tracked in FLOR |
| Bieli et al. (2019) - 1 | 47% | CPS, storms tracked in JRA-55 |
| Bieli et al. (2019) - 2 | 54% | CPS, storms tracked in ERA-Interim |

**Table R1**: ET ratios in the scientific literature

We have clarified this point in the manuscript section 2.8.

**L292 - Is the latitude of minimum pressure dependent on ET? Do you count the minimum post ET or only prior?**

For each storm track, we identified the latitude corresponding to the minimum pressure. Figure 4a (on the left-hand side) shows the latitudes of minimum pressures for all storms while Figure 4b (on the right-hand side) isolates only those that undergo a transition. As a result, the latitude of minimum pressure is not inherently dependent on ET: it can occur either before or after transition. The goal of this analysis was to determine whether storms that undergo ET reach their deepest pressure level further north than those which do not.

We have clarified this point in the manuscript. It now reads:

*"Therefore, the latitude of the minimum pressure is not inherently dependent on ET: it can occur either before or after transition."*

**Figs 2,3,4 - It would be good to also include IBTrACS on these figures as you did with figure 1, to give some idea of how close the model is to these observations (accepting that they can be biased)**

Thank you for this excellent suggestion. To avoid overcrowding the main figures, we created instead an Appendix to further compare IBTrACS to model simulations. The Appendix comprises five additional figures and two additional tables. We also added a short analysis for each of the latter figures and tables.

**Fig 5 - The Eady growth rate is shown at 200hPa, but earlier you only describe the calculation of Eady growth rate at 500hPa. Also, it is confusing that you say you use data at 400 and 500hPa to get the Eady growth rate at 500hPa. Would this not be better described as the Eady growth rate over that layer or at 450hPa assuming you are using 1st order differences.**

There is little literature that provides an explicit formula for the calculation of the Eady growth rate.

The calculation of the Eady growth rate involves computing the first derivatives of the wind velocity and the potential temperature. For the Eady Growth rate at 500 hPa, we use a forward difference scheme using the 400 hPa and 500 hPa values.

Similarly, for the Eady Growth Rate at 200 hPa - where our goal is to assess the baroclinicity in the upper troposphere - we use a backward difference scheme using the 300 hPa and 200 hPa values. A forward scheme in this case would have required using the 100 hPa values, introducing stratospheric influences, which we aimed to avoid.

We have clarified the computations of the Eady Growth Rate in the manuscript. It now reads:

"*In this study, we mainly focused on mid-troposphere baroclinicity and, therefore, computed the EGR at 500 hPa with a forward scheme, using the geopotential heights, humidity, meridional and zonal wind speeds, and temperatures at 400 hPa and 500 hPa*"

"*To assess the baroclinicity in the upper troposphere, we computed the EGR at 200 hPa with a backward scheme we use a backward difference scheme using the 300 hPa and 200 hPa values. A forward scheme in this case would have required using the 100 hPa values, introducing stratospheric influences, which we aimed to avoid.*"

**L333 - The description of this weighting is slightly confusing. The monthly TC number is already in the ET ratio, so cancels out in the weighting and you would be left with the number of ET events in that month divided by the total number of TCs in the year. Is that correct?**

Yes, you are indeed correct. We have changed the manuscript accordingly and it now reads:

"*This indicator is calculated as follows: for each year, the ET ratio is the number of ET events divided by the total number of TCs and then averaged over 30 years.*"

**L352 - "Indeed, TCs that are most likely to undergo ET need to sustain a minimum energy level at middle latitudes". This sounds reasonable, but I was wondering if anyone has actually shown this. Can you add a reference?**

We have added the reference of Hart & Evans (2001). Indeed, they stated "This suggests that tropical systems that are more intense in the tropical phase are able to survive for a greater period of time in the non-supportive region (between 960-hPa MPI and σ=0.25). Weaker tropical systems are able to intensify after transition, if they can quickly enter a supportive baroclinic environment after leaving the unsupportive tropical environment." They also proved this assertion when analyzing the seasonal cycle of ETs. Indeed, they

found that a high distance between the 960-hPa MPI and a baroclinic zone explains the decreased transition probability found in June and July. On the contrary, transitions probabilities in August-September are found higher and can be partly explained by a lower distance between the 960-hPa MPI line and the baroclinic zone.

**Fig 10 - Can you add some indication of statistical significance to the pattern shown, either to the figure or the discussion**

This figure was obtained using the following methodology: for each grid cell, the algorithm calculated the number of ET onsets (over 30 years) occurring in this specific grid cell relative to the total number of ET onsets in the entire spatial domain (over 30 years). Then, spatial smoothing was applied using a Gaussian kernel. Since this method produces only one value per grid cell for both present-day and future-climate simulations — each already incorporating the full spatial and temporal information—it does not provide multiple independent samples that would allow for standard statistical significance testing. As a result, conventional statistical tests cannot be applied to assess the significance of the patterns shown in the figure.

**L383 - low transitioning -> slow transitioning?**

Thanks for pointing it out, we have now corrected it.

**L298 and L408 - You have used the term "available potential energy" interchangeably with "eady growth rate". While they are related, they are not the same. It would be better just to say Eady growth rate in the text as that is what is shown in the figures.**

Yes, we have modified the manuscript accordingly to address your comment and use Eady growth rate instead of available potential energy.

**L457 - Decrease -> weakening**

Thank you for the suggestion. We changed it as you suggested.

**L480 - I think this paragraph could be split into multiple paragraphs. It's quite long and does discuss different things.**

Thank you for pointing this out, we have modified the manuscript accordingly to address your comment.

**The code/data availability needs improvement. You could just write something about where the model data is stored, but I think the tracks you have generated should be made openly available which should be easy enough to do with zenodo. Similarly the code for the data analysis and figures could be uploaded to a zenodo repository.**

Upon acceptance, we can provide the TC tracks and the code and data for most figures on a zenodo repository.